# An Intelligent Agentic System for Complex Image Restoration Problems

**Kaiwen Zhu**[1,2] [\*][†]  **Jinjin Gu**[3] [†]  **Zhiyuan You**[4,5]  **Yu Qiao**[2]  **Chao Dong**[5,2] [‡]

[1]Shanghai Jiao Tong University  [2]Shanghai Artificial Intelligence Laboratory
[3]The University of Sydney  [4]The Chinese University of Hong Kong
[5]Shenzhen Institutes of Advanced Technology, Chinese Academy of Sciences
`zhukaiwen@pjlab.org.cn, jinjin.gu@sydney.edu.au, zhiyuanyou@foxmail.com,`
`qiaoyu@pjlab.org.cn, chao.dong@siat.ac.cn`

## ABSTRACT

Real-world image restoration (IR) is inherently complex and often requires combining multiple specialized models to address diverse degradations. Inspired by human problem-solving, we propose AgenticIR, an agentic system that mimics the human approach to image processing by following five key stages: *Perception*, *Scheduling*, *Execution*, *Reflection*, and *Rescheduling*. AgenticIR leverages large language models (LLMs) and vision-language models (VLMs) that interact via text generation to dynamically operate a toolbox of IR models. We fine-tune VLMs for image quality analysis and employ LLMs for reasoning, guiding the system step by step. To compensate for LLMs' lack of specific IR knowledge and experience, we introduce a self-exploration method, allowing the LLM to observe and summarize restoration results into referenceable documents. Experiments demonstrate AgenticIR's potential in handling complex IR tasks, representing a promising path toward achieving general intelligence in visual processing. The code is available at `https://github.com/Kaiwen-Zhu/AgenticIR`.

## 1 INTRODUCTION

Image restoration (IR) problems in real-world scenarios are inherently complex. Over the past decades, researchers have abstracted various degradation phenomena into independent IR tasks, proposing advanced models tailored to address these problems. However, in practice, these models are seldom used in isolation. Instead, they serve as optional tools that collaborate with other methods to solve complex problems. For example, today's most successful image processing softwares usually integrate multiple models, allowing users to analyze the image, select models, and interact with the processing. This approach enables users to accomplish tasks far more complex than what any single model could achieve, even when using the most basic processing operations. To advance beyond existing models designed for passive and structured IR tasks, we propose a systematic method capable of assuming a dynamic and agentic role in diverse and complex IR scenarios. We aim for this system to function like a human, assessing the image, selecting, planning, and executing various existing IR models to address challenges that individual models alone struggle to solve. We believe this could be one of the promising paths towards general intelligence in IR.

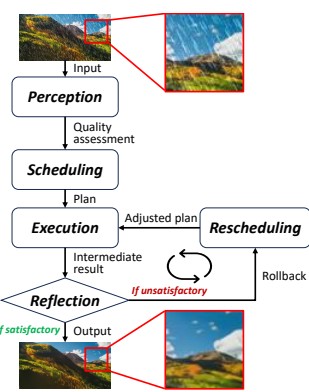

Figure 1: The five stages of human process of IR (some details are hidden).

To achieve this goal, we first need to abstract and generalize how human users use tools to handle IR tasks in real-world scenarios. Fig. 1 illustrates an example. When faced with such an input image, human users may first analyze its quality and degradations (the *Perception* stage), realizing that it contains two types of degradation: noise and rain streaks. Based on the analysis and professional

---

[\*]This work was done during his internship at Shanghai Artificial Intelligence Laboratory.
[†]Equal contribution. [‡]Corresponding author.

experience, they may devise a plan for using tools (the *Scheduling* stage) – for instance, applying a denoising model before a deraining model to prevent noise from affecting the deraining effect. Following the plan, they sequentially apply these models to the image step by step (the *Execution* stage). However, the effects of the models can be dynamic and unpredictable; even models with the same function may behave differently, making it difficult to determine the optimal plan directly. Typically, after an operation, human users may re-analyze the image to assess whether the tool was effective (the *Reflection* stage). If the tool was ineffective or even worsened the image, they undo the operation and formulate a new plan to achieve better results (the *Rescheduling* stage). After repeating this process multiple times as shown in Fig. 1, they eventually achieve a satisfactory result and complete the task. Thus, we abstract the human process of using tools for IR into five stages: *Perception*, *Scheduling*, *Execution*, *Reflection*, and *Rescheduling*. In this work, we propose a system that operates in this manner to tackle tasks in complex IR scenarios.

Constructing such a system is challenging because each of the aforementioned stages requires non-trivial capabilities. We summarize these capabilities as follows:

- *Ability to Analyze Image Quality*: The system needs to understand and assess the quality of images generated at each stage of the processing pipeline. This involves not just outputting a quality score but also understanding the condition of the image quality and identifying the causes of degradation and quality issues. This capability forms the foundation for the perception and reflection stages and provides the necessary information for the (re)scheduling stage.
- *Ability to Reason Based on Context*: The system involves many conditional judgments with complex and ambiguous contexts. Rule-based systems struggle to perform such complex and dynamic reasoning. This ability is crucial for connecting the various stages within the system.
- *Knowledge and Experience in IR*: The system needs to propose a model usage plan based on the condition of the image, which requires the model to have prior knowledge of the behavior of IR models and experience in using them. The system even needs to know the impact of various IR models on each other when they are used in combination.

Fortunately, large language models (LLMs) (OpenAI, 2023a; Touvron et al., 2023) and vision-language models (VLMs) (OpenAI, 2023b; Yin et al., 2023; Liu et al., 2023) offer solutions to these challenges. VLM-based image quality analysis methods provide excellent tools for recognizing complex image degradations, quality assessment, quality comparison, and reasoning about image quality (Wu et al., 2024a;b;c; You et al., 2024b;a). LLMs can perform reasoning and judgment involving complex contexts through text generation. Moreover, advanced pre-trained language models inherently contain foundational knowledge about image restoration, and it is also possible to inject additional prior information into them. The combined and interactive use of LLMs and VLMs can build the IR intelligent agent system we envision.

In this work, we first construct a "toolbox" comprising several IR models and synthetic complex mixed degradation scenarios to emulate real-world problems, serving as the foundational platform for our research. Building upon this, we propose AgenticIR, a system composed of LLMs and VLMs that interact and reason through text generation to dynamically operate the tools within the "toolbox" to solve complex IR tasks. To enable AgenticIR to analyze image quality on demand, we extend a VLM-based image quality assessment method named DepictQA (You et al., 2024a) by fine-tuning it to meet our complex requirement. We also utilize advanced LLMs for reasoning, guiding the system to solve complex problems following the process described in Fig. 1. Although advanced LLMs typically contain some foundational knowledge of IR, they lack understanding of the complex behaviors of IR models and the intricate interactions between models. Therefore, we design a self-exploration and experience summarization method, allowing the LLM to observe and summarize a large number of restoration results, organizing the necessary experiential knowledge into referenceable documents. During the decision-making process, AgenticIR retrieves relevant information and knowledge as concrete ground to make informed decisions.

Our experiments demonstrate the potential of AgenticIR in solving real-world problems. Although our research was primarily conducted in a laboratory environment, we anticipate that this paradigm holds significant promise for applications in automated and intelligent image processing. In practical scenarios, the agents' operations are not limited to individual models; they may involve manipulating different parameters of complex large-scale models (Yu et al., 2024), as well as handling highly dynamic tasks that require judgment and action. We hope that our work can serve as a stepping stone to inspire research toward truly general and intelligent visual processing AI systems.

## 2 RELATED WORK

**Image Restoration (IR)** aims to reconstruct high-quality images from their degraded counterparts. Over the years, this field has witnessed lots of successful models for single-degradation restoration, *e.g.*, denoising (Chen et al., 2023a), deblurring (Nah et al., 2017; Chen et al., 2023c), deraining (Fu et al., 2017; Gu et al., 2023) and super-resolution (SR) (Dong et al., 2016; Ledig et al., 2017; Tao et al., 2024). However, these models' effectiveness is limited to a narrow range as they just focus on specific degradation. There are also efforts devoted to unifying multiple IR tasks (Liu et al., 2024; Chen et al., 2024d). Most of them either mix different degradations into training data (Zhang et al., 2021; Valanarasu et al., 2022; Yu et al., 2024), or predict the degradation type to assist restoration (Li et al., 2022; Luo et al., 2024; Zhang et al., 2023a; Gu et al., 2019; Zhang et al., 2023b; Chen et al., 2024b). Despite remarkable achievements, they may suffer from limitations including optimization difficulty, parameter inefficiency, and poor extensibility. Besides, Yu et al. (2018) and Chen et al. (2024a) invoke multiple single-degradation restoration tools to address multiple degradations and thus are somewhat similar to our work. But Yu et al. (2018) employ reinforcement learning (RL) while Chen et al. (2024a) fine-tune a VLM to directly give an execution plan, not so agentic as ours. Besides, their preoccupation is not combining existing tools as our work.

**LLMs and VLMs** have been at the forefront of advancing general intelligence. Despite being trained on extensive text datasets, their remarkable problem-solving capabilities extend far beyond typical language processing tasks (OpenAI, 2023a;b; Touvron et al., 2023; Liu et al., 2023). LLMs are now capable of handling complex challenges once thought to require human expertise or specialized algorithms, such as mathematical reasoning (Ahn et al., 2024), code generation (Jiang et al., 2024a), and addressing complex legal queries (Cui et al., 2024). Recent research also suggests that LLMs can generate intricate plans for robotics (Driess et al., 2023) and game AI (Zhu et al., 2023), representing a significant step toward their role as general intelligent agents.

**Agents** can be described as systems interacting with environments to solve complicated problems under their agendas (Franklin & Graesser, 1997; Li et al., 2024; Xi et al., 2023; Sumers et al., 2024). Early exploration mainly focuses on symbolic architectures (Franklin & Graesser, 1997) reminiscent of knowledge-based expert systems, rich in expressiveness but poor in flexibility. In the past decade, RL agents have gained success in many tasks (Silver et al., 2018; Hwangbo et al., 2019), but they lack explicitly maintained long-term plans, just acting step by step. LLMs bring new opportunities to this field. With extensive general knowledge, LLMs can answer simple questions logically and flexibly (OpenAI, 2023a). Such responses can function as intermediate steps in a compound problem-solving system. Yao et al. (2023) compare LLMs' response to humans' automatic thinking mode "System 1", while the compound system corresponds to the deliberate thinking mode "System 2" (Kahneman, 2011). Along this trend, many works contextualize LLMs within compound systems to construct language agents. To adapt them to specific tasks and fully harness LLMs' potential, these works draw inspiration from how humans solve problems and intuitively design various mechanisms (Wang et al., 2024). For instance, Tree of Thoughts (Yao et al., 2023) and Graph of Thoughts (Besta et al., 2024) prompt the LLM to provide heuristics for search; HuggingGPT (Shen et al., 2023) and Visual ChatGPT (Wu et al., 2023a) invokes various external tools for diverse tasks; Ghost in the Minecraft (Zhu et al., 2023) and Generative Agents (Park et al., 2023) retrieves from memory to help decision-making; Reflexion (Shinn et al., 2023) reflects on errors in exploration to accumulate experience. As well as foundation models, such compound agent systems integrated with LLMs are widely deemed promising for artificial general intelligence (Zaharia et al., 2024).

## 3 METHOD

We first outline the abstraction of complex IR problems and the design of our research platform (Sec. 3.1). We establish a simulated playground where our AgenticIR can experiment and demonstrate its capabilities. Next, we provide an overview of the workflow design for our proposed method (Sec. 3.2). Finally, we introduce targeted enhancements and designs to equip our system with the specific abilities and knowledge required for effective performance (Sec. 3.3).

### 3.1 RESEARCH PLATFORM DESIGN

We abstract real-world IR scenarios into a set of mixed scenes with well-defined degradations (Kong et al., 2024b; Wang et al., 2021; Kong et al., 2022). While this approach may not cover all possible scenarios, it provides a simple yet general and feasible playground for our research. Specifically, we

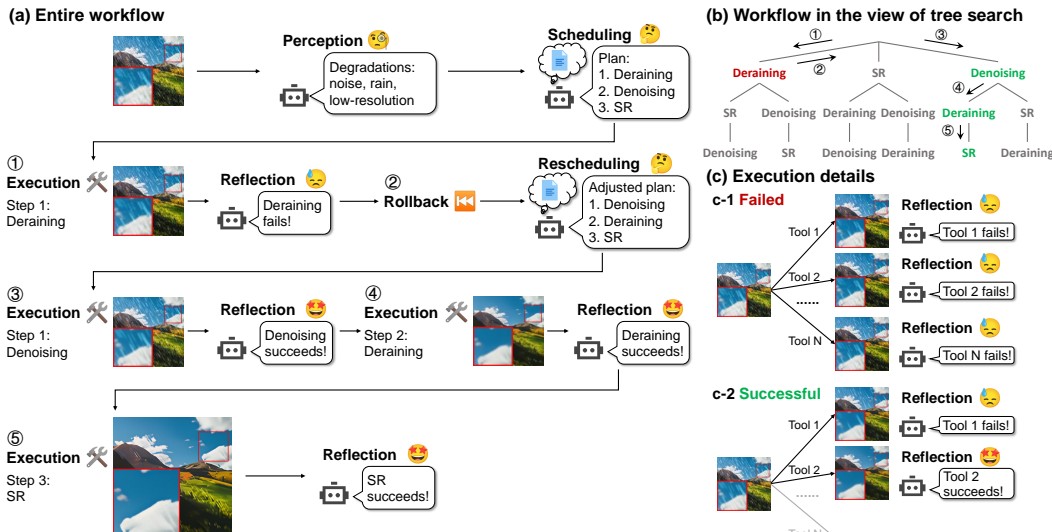

Figure 2: An example illustrating the framework of our AgenticIR. (a) presents the entire workflow, where bubble frames beside robots represent responses from LLMs and VLMs, and the numbers in circles correspond to those in (b). (b) points out the tree search nature of the system. (c) expounds how to execute a single-degradation restoration operation with a toolbox.

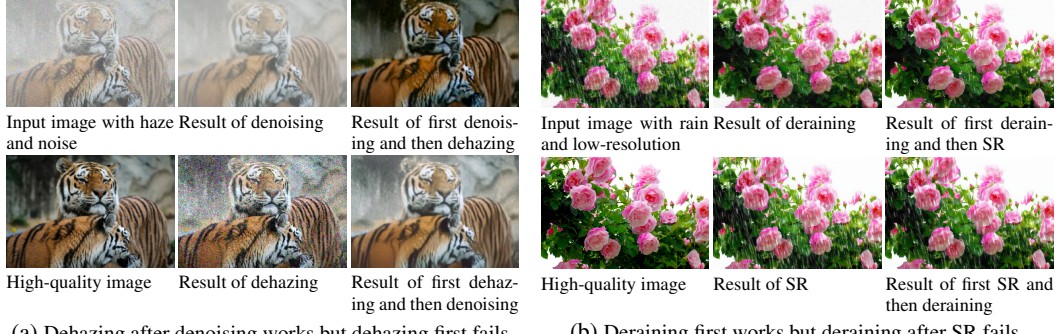

(a) Dehazing after denoising works but dehazing first fails.   (b) Deraining first works but deraining after SR fails.

Figure 3: The importance of operation order in image restoration.

select eight types of degradation: low resolution, noise, motion blur, defocus blur, rain, haze, JPEG compression artifacts, and low light. There are many specialized models for each type of degradation that can be used as tools. For each degradation, we collect three to six advanced models to build the "toolbox" that the intelligent agent can use. Although these models are designed for single-degradation tasks, our goal is for the intelligent agent to leverage them to tackle more complex IR problems. We further create complex restoration tasks of varying difficulty by combining different degradations (Kong et al., 2024b), allowing us to evaluate the agent's capability in solving complex problems and its generalization ability to unseen scenarios. The degraded data used for testing are introduced in the experimental section. More details can be found in the Appendix A.3 and A.4.

## 3.2 WORKFLOW DESIGN

**Overall Workflow.** As illustrated in Fig. 2(a), our agent restores images with complex degradations through a multi-stage process. The agent begins with the *Perception* stage, dynamically analyzing the content and degradations of the input image using a multi-modal vision language model. This information is then utilized in the subsequent *Scheduling* stage by the language model. In the *Scheduling* stage, the language model leverages the identified degradation information, its inherent commonsense knowledge about degradation restoration, and additional domain-specific knowledge provided by us to formulate judgments and develop a possible overall plan to restore the image. This plan consists of a sequence of operations. Following this, in the *Execution* stage, the agent executes the first operation in the plan, as shown in Fig. 2(c). The agent incrementally tries available tools in an attempt to achieve the goal of the plan. During this process, the *Reflection* mechanism evaluates whether a tool has been successful and whether an operation has achieved its intended purpose. In

our approach, reflection is also implemented by a multi-modal model with degradation and quality awareness. If the goal is met, the agent proceeds to the next operation, repeating this cycle until all operations are completed and the restoration is finalized; otherwise, rollback is triggered (introduced later). For more details, please refer to Appendix A.1.

**The Importance of Scheduling.** In our workflow, correct scheduling is crucial because the complexity of IR model behaviors means that the selection of models and their execution order can greatly impact the final outcome. In real-world scenarios, complex degradations are mixtures of multiple degradations, and the mutual influence between different models leads to fundamental differences in results based on model selection and execution sequence. Fig. 3 illustrates two examples. In the first example, due to the simultaneous presence of haze and noise, if we directly use a dehazing model, the noise interferes with the dehazing model's ability, resulting in failure. In the second example, the original rain streaks can be removed by a deraining model, but after super-resolution processing, the rain streaks exceed the deraining model's processing capabilities. The main purpose of the *Scheduling* stage is to develop better IR strategies for the current task and all the phenomena described above need to be taken into account.

**Rollback Mechanism.** However, the plan may encounter issues, as the scheduling stage is not guaranteed to give the optimal strategy. If execution at any stage fails to meet its objective, we have the reflection mechanism to identify the failure. Inspired by human interaction with image processing software, we design a rollback mechanism where the agent returns to the previous stage, learns from the failure, and creates a new plan (*Rescheduling*). The agent then re-enters the execution stage with the updated plan. This process essentially performs a depth-first tree search among all possible degradation plans, as shown in Fig. 2(b). Given the vast number of degradation possibilities and the complexity of the situations, it is impractical to exhaustively traverse all options in real-world applications. However, our method provides an efficient mechanism to find feasible restoration paths. The LLM's reasoning capabilities, its knowledge of IR, and the experiential information we supply all contribute to improving the efficiency of this search and decision-making process.

### 3.3 CAPABILITY ACQUISITION

Sec. 3.2 outlined the workflow of the proposed AgenticIR. To make this complex process both feasible and effective, the system requires several non-trivial capabilities. In the following, we describe these essential capabilities and how we equip the system to acquire them.

**Ability to Analyze Image Quality.** For an agentic system that processes images based on varying degradation conditions, it is crucial to provide it with the ability to recognize, analyze, and assess image quality. While numerous models exist for image quality assessment, most simply output scores or similarities. These approaches are insufficient for two key reasons: (1) they fail to provide the necessary information for the subsequent scheduling stage; and (2) they cannot establish a threshold to judge the success of an execution, as a mere score does not indicate whether a specific degradation has been effectively removed. To address this limitation, we focus on a new form of image quality perception and assessment – one based on VLMs (You et al., 2024b;a; Wu et al., 2024a;b;d). These models can describe both the content and quality issues of an image using natural language, offering critical information for the scheduling stage to develop a restoration plan. Some VLMs can even perform reasoning and answer questions, enabling accurate judgment of the success or failure of specific executions. Our system incorporates a VLM-based image quality assessment model to acquire these advanced capabilities.

Specifically, we extend DepictQA (You et al., 2024a) – a successful multi-modal image quality assessment model. DepictQA, trained on a large dataset, can analyze image content and categorize types of quality issues, evaluate the quality of single images, and compare the quality of multiple images. It already possesses substantial knowledge and capabilities in image quality assessment. We further adapt DepictQA to the specific requirements of our system through LoRA fine-tuning (Hu et al., 2022). We additionally train the model to evaluate the severity of various degradations in images. With this extension, the DepictQA method can evaluate each plan execution effectively. Implementation details are described in Appendix A.2.

**Ability to Reason Based on Context.** To effectively connect the various stages within the agent system – especially during the scheduling and rescheduling stages – the agent must have a thorough understanding of the tasks and workflow. In such a dynamic and complex system, making judgments based on predefined rules is nearly impractical. Moreover, the increasing number of pos-

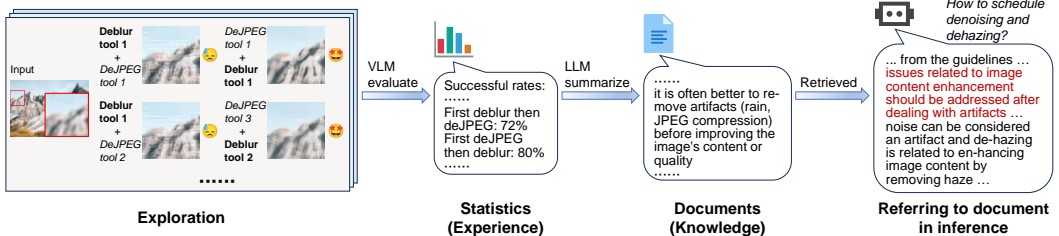

Figure 4: LLMs alone fail to grasp the intricate interactions among operations and thus cannot plan reliably. To address it, we let the agent self-explore beforehand and then summarize the accumulated experience to distill knowledge. The knowledge will be a concrete ground for planning in inference.

Table 1: Prompting GPT-4 (OpenAI, 2023a) to plan.

| ID | Prompt | Response | |
|----|--------|----------|---|
| 1 | ... we will conduct these tasks: ['motion deblurring', 'jpeg compression artifact removal']. Please provide some insights into the correct order of these unordered tasks. | ... Motion deblurring should ideally be performed first because it works best on images that have not been subjected to additional processing ... | ✗ |
| 2 | ... we will conduct these tasks: ['jpeg compression artifact removal', 'motion deblurring']. Please provide some insights into the correct order of these unordered tasks. | ... deblurring itself may exacerbate the appearance of compression artifacts, as it attempts to sharpen the image and restore detail ... | ✗ |
| 3 | ... we will conduct these tasks: ['motion deblurring', 'jpeg compression artifact removal'] ... we have the following experience ... please give the correct order of the tasks. | ... the experience suggests that it is often better to remove artifacts before improving the image's content or quality ... we should remove JPEG artifacts before addressing the motion blur. | ✓ |

sible states and heightened complexity make rule-based approaches exceedingly convoluted, and many of the judgment conditions are ambiguous. This necessitates that the agent system possesses a considerable degree of intelligent "thinking" and "reasoning" capabilities. We utilize advanced LLMs to perform this reasoning, serving as a bridge that links the different stages of the system and facilitates thought processes. Trained on vast amounts of human language data, LLMs can mimic human reasoning through text generation. The reasoning abilities of LLMs have been recognized across various tasks (OpenAI, 2023a; Wei et al., 2022; Zhou et al., 2023). In our work, we employ GPT-4 (OpenAI, 2023a) for this reasoning because it possesses excellent comprehensive abilities and knowledge that can be conveniently accessed. It is important to note that other language models with sufficiently strong capabilities can also be used to construct this agent system.

**Knowledge and Experience in IR.** As previously discussed, the execution order of IR models significantly impacts the final results. We rely on the capabilities of LLMs to formulate model usage plans based on image conditions. While LLMs are extensively trained and possess some knowledge of IR, this foundational knowledge alone is insufficient to master the usage experience of the diverse tools in our toolbox. For example, in Cases 1 and 2 of Tab. 1, we prompt GPT-4 (OpenAI, 2023a) to arrange the sequence of motion deblurring and JPEG artifact removal. When we change the presentation order, which is meaningless, of these two degradations, GPT-4's answers also change. This illustrates that GPT-4 cannot reliably resolve these issues because it is merely speculating. This echoes the viewpoint of Kambhampati et al. (2024): LLMs possess only general planning knowledge but cannot truly handle subtask interactions, thus failing to provide executable plans.

To address this limitation, we need to inform the LLM of experiential information about the usage of tools in the toolbox during reasoning, allowing it to base its reasoning on the prior knowledge we provide. To obtain comprehensive prior knowledge and minimize human subjective judgment, we propose a method of self-exploration and summarization, as illustrated in Fig. 4. The agent first actively explores actual scenarios to accumulate experience and summarize this experience into referenceable documents for future reasoning. Specifically, for some images with complex degradations, we apply all possible sequences of the corresponding tools and use the fine-tuned DepictQA to evaluate the restored images, determining whether the restoration is successful. We calculate the success rate of each operation sequence. After collecting these statistics, we prompt GPT-4 to summarize them and store the results in a knowledge base, which will be retrieved during reasoning to assist in operation scheduling. For example, when we prompt GPT-4 again to arrange

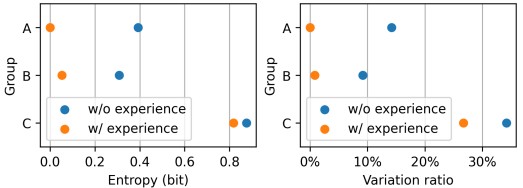

Figure 5: Comparison between dispersion of scheduling results with and without experience. Lower metric indicates higher consistency.

Table 2: Degradation evaluation performance of fine-tuned DepictQA.

| Degradation | Precision | Recall | F1 score |
|---|---|---|---|
| Noise | 0.99 | 0.92 | 0.95 |
| Motion blur | 0.88 | 0.52 | 0.65 |
| Defocus blur | 0.82 | 0.65 | 0.72 |
| JPEG compression artifact | 0.98 | 1.00 | 0.99 |
| Rain | 0.97 | 0.98 | 0.98 |
| Haze | 0.88 | 0.91 | 0.89 |
| Low light | 0.87 | 0.65 | 0.74 |

Table 3: Quantitative comparison between acting as the agent's plan and the opposite. "Not as planned" means randomly shuffling the plan (guaranteed to be different).

| Degradations | As planned | PSNR | SSIM | LPIPS↓ | MANIQA | CLIP-IQA | MUSIQ |
|---|---|---|---|---|---|---|---|
| Group A | ✓ | **21.14** | **0.6836** | **0.2753** | 0.3469 | **0.5091** | **60.77** |
| | ✗ | 20.79 | 0.6652 | 0.3060 | 0.3385 | 0.4819 | 59.85 |
| Group B | ✓ | **21.14** | **0.7088** | **0.2683** | 0.3588 | **0.5275** | **61.92** |
| | ✗ | 20.32 | 0.6811 | 0.2976 | **0.3623** | 0.5257 | 60.15 |
| Group C | ✓ | **18.78** | **0.5352** | **0.4239** | **0.3118** | **0.4876** | 51.08 |
| | ✗ | 18.49 | 0.5277 | 0.4345 | 0.3058 | 0.4719 | **51.32** |

Table 4: Quantitative comparison with the random tool invocation. The better performances are marked in **bold**. ↓ means the lower the better, and for others, the higher the better.

| Degradations | Method | PSNR | SSIM | LPIPS↓ | MANIQA | CLIP-IQA | MUSIQ |
|---|---|---|---|---|---|---|---|
| Group A | AgenticIR | **21.04** | **0.6818** | **0.3148** | **0.3071** | **0.4474** | **56.88** |
| | Random | 20.90 | 0.6642 | 0.3368 | 0.2963 | 0.4394 | 55.30 |
| Group B | AgenticIR | **20.55** | **0.7009** | **0.3072** | **0.3204** | **0.4648** | **57.57** |
| | Random | 20.06 | 0.6766 | 0.3351 | 0.3120 | 0.4514 | 56.15 |
| Group C | AgenticIR | 18.82 | **0.5474** | **0.4493** | **0.2698** | **0.3948** | **48.68** |
| | Random | **18.87** | 0.5456 | 0.4796 | 0.2354 | 0.3543 | 44.61 |

the order of motion deblurring and JPEG artifact removal using this experiential knowledge, it can find references in the experience and consistently provide answers like Response 3 in Tab. 1. In this way, GPT-4 can offer reliable heuristics for the execution order of operations.

## 4 EXPERIMENTS

As described in Sec. 3.1, we construct our research data using mixed degradations. In the experimental section, we first build the test dataset. We designed 16 combinations of mixed degradations involving 2 or 3 types of degradation and divided them into three groups: A, B, and C. Group A contains 8 combinations, while groups B and C each contain 4 combinations. The degradation combinations in groups A and B consist of 2 degradations, whereas those in group C consist of 3 degradations to simulate more complex situations. During the exploration phase, the agent is exposed only to group A – that is, the agent is familiar with the degradations present in group A but is unaware of those in groups B and C. This setup helps us investigate the system's generalization ability. We applied each of the 16 degradation combinations to every one of the 100 images in the MiO100 (Kong et al., 2024a;b) dataset. For each combination in group A, we allocated 20 images for exploration, totaling 160 images. The remaining 1,440 images are used for testing. More detailed information can be found in Appendix A.3.

### 4.1 EFFECTIVENESS OF INDIVIDUAL DESIGNS

**Fine-Tuning DepictQA for Image Quality Analysis.** For each image in the test set, we prompted the fine-tuned DepictQA to assess the severity of each degradation. By treating degradation assessment as a binary classification problem—that is, determining whether an image suffers from a particular degradation—we computed the precision, recall, and F1 scores, as shown in Tab. 2. Considering that we did not invest effort in designing special strategies and that the fine-tuning consumed only moderate computational resources (three hours on four V100 GPUs), the learning process proved to be quite efficient. The fine-tuned DepictQA effectively recognized almost all types of degradation, except that the recall rates for defocus blur and motion blur were somewhat limited. This limitation is understandable, as these two types of blur are very similar in many cases; even humans find it difficult to fully distinguish between them.

**Self-Exploration and Experience Summarization.** Informed of successful rates of each operation order for each degradation combination in group A, GPT-4 tries to summarize them and distill insights. Fig. 9 shows snippets of the distilled insights and how they help operation scheduling. It can be seen that GPT-4 does conclude general rules that seem reasonable and apply them to infer the order of operations logically. The scheduling of "deraining, super-resolution" and "denoising, dehazing" also echo our discussion about Fig. 3 in Sec. 3.3. The example in Fig. 6 shows the correctness of the plans derived from experience: without experience, the agent decides to deblur first, only

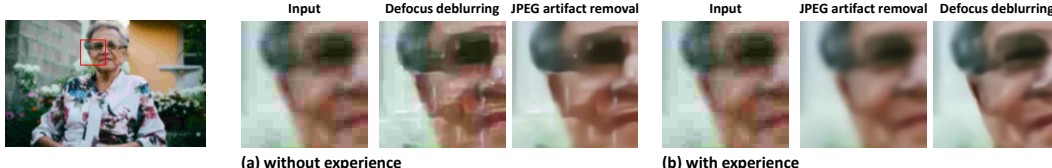

Figure 6: An example showing the correctness of scheduling with experience.

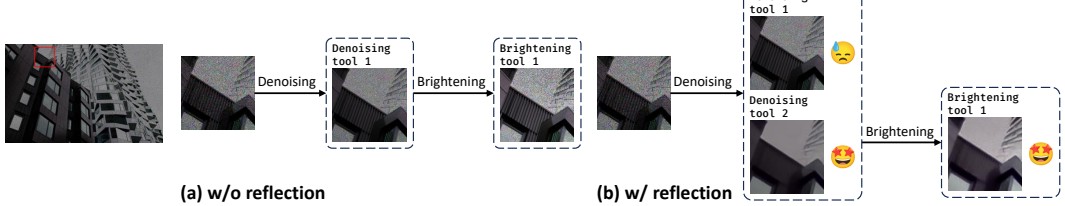

Figure 7: Exemplary comparison between with and without reflection.

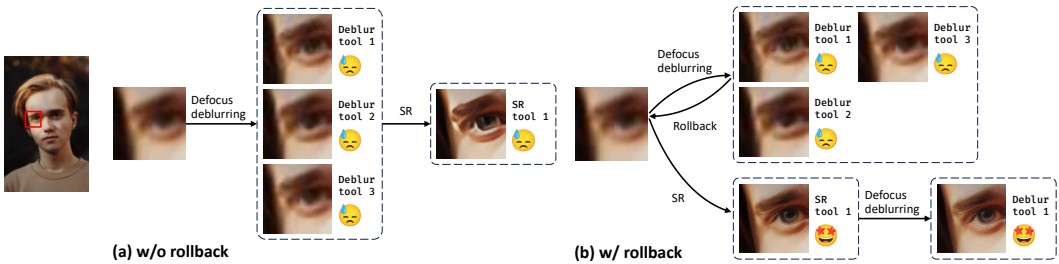

Figure 8: Exemplary comparison between with and without rollback.

to distort the JPEG artifact and render it hard to remove; scheduling with experience can avoid this. To quantitatively evaluate, we let the agent act as the plans to restore images in the test set; as the control group, we conduct another experiment wherein the agent does not act as the plans. Six image quality assessment metrics are used for evaluation: three full-reference metrics PSNR, SSIM (Wang et al., 2004), LPIPS (Zhang et al., 2018), and three non-reference metrics MANIQA (Yang et al., 2022), CLIP-IQA (Wang et al., 2023), MUSIQ (Ke et al., 2021). Tab. 3 lists the results, which indicate the efficacy of the proposed method. For more details, refer to Appendix B.1 and B.2.

We also investigate the consistency improvement to justify the motivation introduced in Sec. 3.3: for a set of operations, GPT-4 fails to consistently give a scheduling result, especially sensitive to the presentation order of operations; providing experience should alleviate this problem. We prompt GPT-4 to schedule each set 60 times with random presentation order, and measure the dispersion of the results. Two metrics are adopted: (1) treating the results as a discrete distribution over all permutations of operations, *entropy* reflects the overall dispersion; (2) *variation ratio* is the proportion of samples that are not mode, straightforwardly reflecting confidence in the dominant result. Fig. 5 compares the dispersion of scheduling results of the 16 operation sets with and without experience averaged in groups. In fact, for some operation sets, the experience improves the consistency from almost random to deterministic (refer to Appendix B.3 for details). These results support our claim.

**Inference Workflow.** There are two mechanisms in the inference workflow to ablate: reflection, *i.e.*, whether to check the tool results, and rollback, *i.e.*, whether to roll back failed operations. We compare the performances with and without the two mechanisms respectively[1], and the quantitative results[2] are listed in Tab. 5. We can see the absence of any mechanism leads to a performance drop in most metrics. Qualitative comparisons are shown in Fig. 7 and Fig. 8. In Fig. 7, without reflection, although the plans are the same, the agent fails to select the appropriate tool, resulting in remnant noise. In Fig. 8, without rollback, the agent fails to correct the suboptimal operation order

---

[1]Reflection is a prerequisite of rollback, so rollback is disabled when ablating reflection.

[2]When ablating rollback, to highlight the impact, the statistics are only calculated on cases wherein rollback is triggered (20% cases).

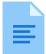

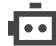

**Insights distilled from experience**
… From these observations, we can infer that generally ... it is often better to remove artifacts (rain, JPEG compression) before improving the image's content or quality ...

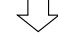 *Guide: How to schedule ... ?*

**Deraining, SR (in group A)**
According to the collected experience … it is more effective to address the artifact caused by rain before enhancing the image's resolution ...

**Denoising, dehazing (in group B)**
... from the guidelines … issues related to image content enhancement should be addressed after dealing with artifacts … noise can be considered an artifact and dehazing is related to enhancing image content by removing haze …

**Brightening, defocus deblurring, JPEG artifact removal (in group C)**
... more effective to address blurring issues before image content enhancement ... removing artifacts like JPEG compression should be done before addressing blurring issues …

Figure 9: Examples of GPT-4's responses for operation scheduling.

Table 5: Ablation studies of the inference workflow, including reflection (Ref.) and rollback (Rb.).

(a) Ablation study of reflection.

| Degradations | Ref. | Rb. | PSNR | SSIM | LPIPS↓ | MANIQA | CLIP-IQA | MUSIQ |
|---|---|---|---|---|---|---|---|---|
| Group A | ✓ | ✗ | **21.12** | **0.6809** | 0.3079 | **0.3179** | **0.4617** | **57.52** |
| | ✗ | ✗ | 20.47 | 0.6659 | 0.3282 | 0.2906 | 0.4387 | 55.56 |
| Group B | ✓ | ✗ | **20.74** | **0.6986** | 0.3084 | **0.3126** | **0.4567** | **56.66** |
| | ✗ | ✗ | 20.46 | 0.6798 | 0.3412 | 0.2966 | 0.4359 | 54.81 |
| Group C | ✓ | ✗ | 18.85 | **0.5510** | **0.4559** | **0.2557** | **0.3771** | **47.38** |
| | ✗ | ✗ | **18.93** | 0.5447 | 0.4764 | 0.2349 | 0.3595 | 43.77 |

(b) Ablation study of rollback.

| Degradations | Ref. | Rb. | PSNR | SSIM | LPIPS↓ | MANIQA | CLIP-IQA | MUSIQ |
|---|---|---|---|---|---|---|---|---|
| Group A | ✓ | ✓ | **20.23** | 0.6626 | 0.3249 | **0.3197** | 0.4158 | **59.87** |
| | ✓ | ✗ | 19.77 | **0.6725** | **0.3067** | 0.3042 | **0.4484** | 58.70 |
| Group B | ✓ | ✓ | **18.76** | **0.6642** | 0.3348 | **0.3251** | 0.4525 | **57.43** |
| | ✓ | ✗ | 18.30 | 0.6348 | 0.3591 | 0.3082 | **0.4528** | 55.94 |
| Group C | ✓ | ✓ | **18.99** | **0.5461** | **0.4604** | **0.2643** | **0.3974** | **49.64** |
| | ✓ | ✗ | 18.64 | 0.5446 | 0.4634 | 0.2348 | 0.3669 | 46.73 |

suggested by the experience: although defocus deblurring before SR makes sense, the defocus blur in this case is slight enough to be addressed by SR tools[3], but first conducting defocus deblurring over-smoothens the image. These examples show that the mechanisms are necessary in various scenarios, working together to enable the agent to find the proper operation order and tools.

## 4.2 COMPARISON WITH OTHER METHODS

We test AgenticIR on our test set for the complex-degradation restoration task. To our knowledge, there is no open-source work for IR tool ensemble yet, so we design a simple method to compare: with the degradations predicted by our fine-tuned DepictQA, randomly selecting one tool for each degradation and invoking them in random order. Tab. 4 lists the quantitative results. AgenticIR outperforms this method in almost all metrics.

As for the IR performance, we compare AgenticIR with several all-in-one models: AirNet (Li et al., 2022), PromptIR (Potlapalli et al., 2023), MiOIR (Kong et al., 2024a), DA-CLIP (Luo et al., 2024), InstructIR (Conde et al., 2024), and AutoDIR (Jiang et al., 2024b). Tab. 6 lists the quantitative comparison. AgenticIR

Table 6: Quantitative comparison with all-in-one models. The best and second best performances are marked in **bold** and underline respectively.

| Degradations | Method | PSNR | SSIM | LPIPS↓ | MANIQA | CLIP-IQA | MUSIQ |
|---|---|---|---|---|---|---|---|
| Group A | AirNet | 19.13 | 0.6019 | 0.4283 | 0.2581 | 0.3930 | 42.46 |
| | PromptIR | 20.06 | 0.6088 | 0.4127 | 0.2633 | 0.4013 | 42.62 |
| | MiOIR | 20.84 | 0.6558 | 0.3715 | 0.2451 | 0.3933 | 47.82 |
| | DA-CLIP | 19.58 | 0.6032 | 0.4266 | 0.2418 | 0.4139 | 42.51 |
| | InstructIR | 18.03 | 0.5751 | 0.4429 | 0.2660 | 0.3528 | 45.77 |
| | AutoDIR | 19.64 | 0.6286 | 0.3967 | 0.2500 | 0.3767 | 47.01 |
| | AgenticIR | **21.04** | **0.6818** | 0.3148 | **0.3071** | **0.4474** | **56.88** |
| Group B | AirNet | 19.31 | 0.6567 | 0.3670 | 0.2882 | 0.4274 | 47.88 |
| | PromptIR | 20.47 | 0.6704 | 0.3370 | 0.2893 | 0.4289 | 48.10 |
| | MiOIR | 20.56 | 0.6905 | 0.3243 | 0.2638 | 0.4330 | 51.87 |
| | DA-CLIP | 18.56 | 0.5946 | 0.4405 | 0.2435 | 0.4154 | 43.70 |
| | InstructIR | 18.34 | 0.6235 | 0.4072 | 0.3022 | 0.3790 | 50.94 |
| | AutoDIR | 19.90 | 0.6643 | 0.3542 | 0.2534 | 0.3986 | 49.64 |
| | AgenticIR | 20.55 | 0.7009 | 0.3072 | 0.3204 | 0.4648 | **57.57** |
| Group C | AirNet | 17.95 | 0.5145 | 0.5782 | 0.1854 | 0.3113 | 30.12 |
| | PromptIR | 18.51 | 0.5166 | 0.5756 | 0.1906 | 0.3104 | 29.71 |
| | MiOIR | 15.63 | 0.4896 | 0.5376 | 0.1717 | 0.2891 | 37.95 |
| | DA-CLIP | 18.53 | 0.5320 | 0.5335 | 0.1916 | 0.3476 | 33.87 |
| | InstructIR | 17.09 | 0.5135 | 0.5582 | 0.1732 | 0.2537 | 33.69 |
| | AutoDIR | 18.61 | 0.5443 | 0.5019 | 0.2045 | 0.2939 | 37.86 |
| | AgenticIR | 18.82 | 0.5474 | 0.4493 | 0.2698 | 0.3948 | **48.68** |

wins in almost all metrics. Note that by this comparison we do not intend to argue the superiority of AgenticIR over all-in-one models. Instead, we hope to verify that combining single-degradation IR tools can achieve comparable performance with all-in-one models, so this approach is feasible and meaningful. After all, the intrinsic purpose of this paper is not IR but IR tool ensemble.

Qualitative comparison with the baseline and all-in-one models is shown in Fig. 10. We can see AgenticIR effectively addresses all degradations and yields visually pleasing results. In contrast, random invocations may fail to find the appropriate operation order and tools, and all-in-one models may struggle to address all degradations due to their diverse and even conflicting characteristics.

---

[3]The training data of many SR models also include blur, as proposed by Wang et al. (2021).

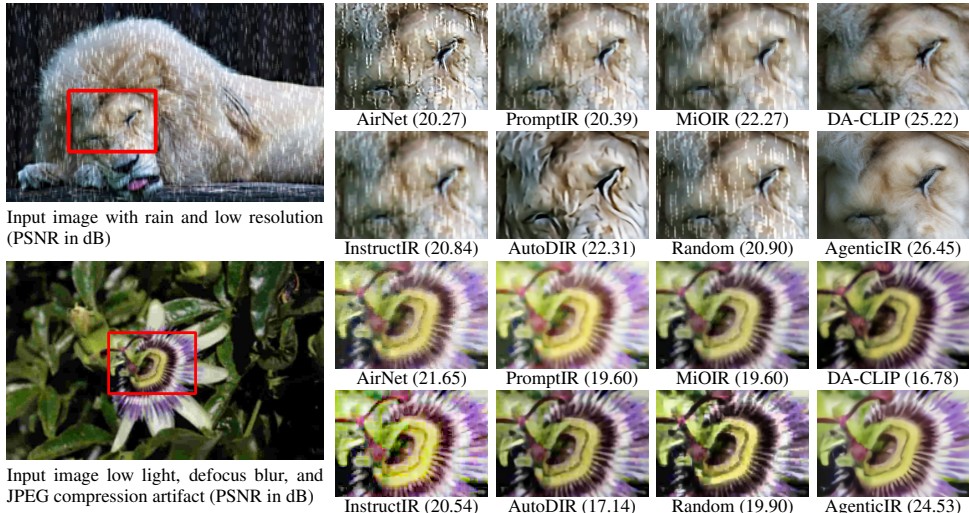

Figure 10: Qualitative comparison with other methods.

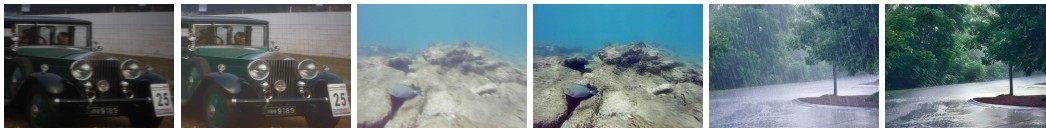

Figure 11: Examples of AgenticIR restoring real-world complexly degraded images. First two images: taken by under-display camera, restored by motion deblurring, defocus deblurring, and brightening; middle two images: taken underwater, restored by defocus deblurring, dehazing, and motion deblurring; last two images: taken in heavy rain, restored by deraining and dehazing.

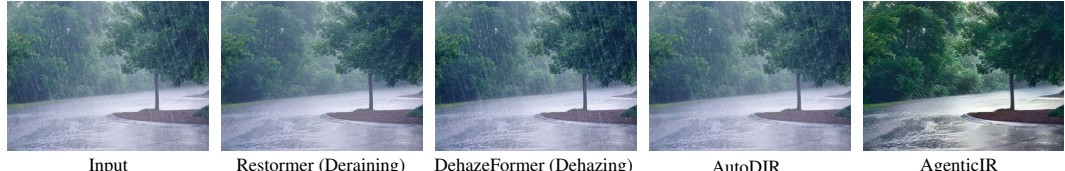

Figure 12: Comparison with individual models on a real-world image.

**Real-World Applications.** We also verify AgenticIR's ability on real-world complexly degraded images. Fig. 11 shows some examples: the first one is from an under-display camera dataset (Zhou et al., 2021), and AgenticIR restores it by motion deblurring, defocus deblurring, and brightening; the second one is from an underwater dataset (Islam et al., 2020), and AgenticIR restores it by defocus deblurring, dehazing, and motion deblurring (similar decomposition is also observed by Jiang et al. (2024b)); the last one is an image with heavy rain downloaded from the Internet, and AgenticIR restores it by deraining and dehazing. These examples illustrated that AgenticIR can really restore some real-world low-quality images by decomposing the complex-degradation IR task into several tractable single-degradation operations, verifying the application value of AgenticIR. Fig. 12 compares AgenticIR with individual models. In this example, the deraining model (Restormer (Zamir et al., 2022)), dehazing model (DehazeFormer (Song et al., 2023)), and all-in-one model (AutoDIR (Jiang et al., 2024b)) all behave poorly, while AgenticIR, sequentially conducting deraining by Restormer and dehazing by DehazeFormer, manages to yield a clean image. This illustrates that AgenticIR as a compound system does have an advantage over individual models on some occasions.

## 5 CONCLUSION

In conclusion, we introduce AgenticIR, a system that combines LLMs and VLMs to emulate human-like strategies in complex IR tasks by dynamically utilizing a toolbox of IR models through the stages of perception, scheduling, execution, reflection, and rescheduling. Our experiments demonstrate that AgenticIR effectively tackles complex IR problems beyond the capability of individual models, showing significant potential for applications in automated and intelligent image processing. We hope this work can lay a promising foundation for future research toward truly general and intelligent visual processing AI systems.

## ACKNOWLEDGEMENTS

This work is supported by the National Key R&D Program of China (No. 2022ZD0160102), the National Natural Science Foundation of China (Grant No. 62276251), and the Joint Lab of CAS-HK.

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

# A  MORE IMPLEMENTATION DETAILS

## A.1  INFERENCE WORKFLOW

**LLM-Implemented Functions.**  Algorithm 1 and 2 describe the entire inference workflow of AgenticIR, where the functions EVALU-ATE, SCHEDULE, REFLECT, RESCHEDULE and PICKBEST are implemented by LLMs. In the function EVALUATE, fine-tuned Depic-tQA (You et al., 2024a) evaluates the sever-ities of all types of degradation to recognize degradations present in the image; if DepictQA thinks the severity is medium, high, or very high, then the degradation is deemed existent. In REFLECT, DepictQA evaluates the degra-dation corresponding to the single-degradation restoration subtask at hand to judge whether the execution is successful. In SCHEDULE and RESCHEDULE, GPT-4 (OpenAI, 2023a) (in this paper we use the `gpt-4-1106-preview` version) references accumulated experience to infer the order of given subtasks; specially, in RESCHEDULE, GPT-4 is informed the previous failed attempts. In PICKBEST, the candidates are linearly scanned and compared in pairs by DepictQA, resulting in the highest quality can-didate. Prompts are listed in Tab. 7.

**Subtask Execution.**  To select the appropriate tool for the subtask, we iteratively invoke and reflect. After invoking a tool, if the fine-tuned DepictQA thinks the severity of the degradation is very low, then we accept the tool result im-mediately; otherwise, we continue to try other tools. If no tool gives very low degradation severity but some give low severity, then we run PICKBEST to select the best one of them as a successful subtask result. If no tool gives low or very low severity, then the subtask is reported as failed; in this case, we still run PICKBEST to find the best one, which is useful if all subtask orders from the input image fail.

---

**Algorithm 1:** Inference workflow

**Input:** Low-quality image $I$
**Output:** Restored high-quality image

1  $agenda \leftarrow$ EVALUATE$(I)$;
2  $plan \leftarrow$ SCHEDULE$(agenda)$;
3  **while** $plan$ is not empty **do**
4  $\quad$ $I, success \leftarrow$ DFS$(I, plan)$;
5  $\quad$ **if** $success$ **then**
6  $\quad\quad$ output $I$;
7  $\quad$ **else**
8  $\quad\quad$ $plan \leftarrow$ the remaining plan for $I$;
9  **output** $I$;

---

**Algorithm 2:** DFS

**Input:** Image $I$, list of subtasks $plan$
**Output:** Restored image, successful or not

1  **if** $plan$ is empty **then**
2  $\quad$ **output** $I$, true;
3  $attempts \leftarrow \emptyset$;
4  $inferiors \leftarrow \emptyset$;
5  **repeat**
6  $\quad$ $subtask \leftarrow$ the first subtask of $plan$;
7  $\quad$ $\tilde{I} \leftarrow$ result of $subtask$ on $I$;
8  $\quad$ $pass \leftarrow$ REFLECT$(\tilde{I}, subtask)$;
9  $\quad$ **if** $pass$ **then**
10  $\quad\quad$ Remove $subtask$ from $plan$;
11  $\quad\quad$ $\tilde{I}, success \leftarrow$ DFS$(\tilde{I}, plan)$;
12  $\quad\quad$ **if** $success$ **then**
13  $\quad\quad\quad$ **output** $\tilde{I}$, true;
14  $\quad$ Add $subtask$ to $attempts$;
15  $\quad$ Add $\tilde{I}$ to $inferiors$;
16  $\quad$ **if** size of $attempts \neq$ size of $plan$ **then**
17  $\quad\quad$ $plan \leftarrow$ RESCHEDULE$(plan, attempts)$;
18  $\quad$ **else**
19  $\quad\quad$ $I \leftarrow$ PICKBEST$(inferiors)$;
20  $\quad\quad$ **output** $I$, false;

---

## A.2  FINE-TUNING VLM

DIV2K (Agustsson & Timofte, 2017) and Flickr2K (Timofte et al., 2017) datasets are used for fine-tuning DepictQA (You et al., 2024a) on degradation evaluation. We randomly add two to four degradations on each image to obtain 15,000 complexly degraded images. For each of them, we iterate the seven degradations[4] to synthesize seven question-answer pairs: question: "What's the severity of `degradation` in this image?"; answer: "very low / low / medium / high / very high". We fine-tune DepictQA for one epoch with batch size 64 on 4 NVIDIA Tesla V100 GPUs, using learning rate 0.0005, weight decay 0.001, and Adam optimizer ($\beta_1 = 0.9, \beta_2 = 0.95$).

---

[4]The degradation "low resolution" is excluded since we can directly recognize it by checking the resolution.

## A.3 DATA

This section details how we synthesize degraded images. We first introduce how the eight degradations are implemented respectively and then list the combinations.

**Single Degradation.** For low resolution, following most SR works (Dong et al., 2016; Liang et al., 2021), we downsample the images by a factor of 4 using bicubic interpolation. For noise, we add Gaussian or Poisson noise with random scale. For motion blur, following Michaelis et al. (2019), we filter images with linear kernels with random direction and radius. For defocus blur, following Michaelis et al. (2019), we filter images with circular kernels with random radius. For rain, following Kong et al. (2024a), we first add noise and then filter the noise with linear kernels with random direction. For haze, following most dehazing works (He et al., 2009; Li et al., 2019), we adopt the atmospheric scattering model with random global atmospheric light and scattering coefficient. For JPEG compression artifact, we compress images with random quality factor. For low light, we randomly reduce the V channel value of HSV color space with one of the following approaches: subtracting a constant, Gamma correction, and linear mapping.

**Degradation Combination.** Tab. 8 lists the 16 degradation combinations for exploration and testing. Note that we consider real-world scenarios deliberately when designing the combinations. The combinations should be common in reality, *e.g.*, haze and rain, dark and noise. Besides, the degradation order should conform to physical limitations, *e.g.*, blur should be added before noise since they follow a chronological order in imaging.

## A.4 TOOLS

Tab. 9 lists the single-degradation restoration tools adopted in our implementation. Except for brightening, all tools are cutting-edge deep models. For brightening, deep models are not employed since the low-light conditions considered here are not so severe as those considered by deep models (Wei et al., 2018) for practical purposes.

Table 7: Prompts for VLM and LLM. {·} represents slot to fill according to the context.

| Prompt for DepictQA to evaluate degradation |
|---|
| ```
What's the severity of {degradation} in this image?  Answer the
question using a single word or phrase in the followings:  very
low, low, medium, high, very high.
``` |
| Prompt for GPT-4 to (re)schedule subtasks |
| ```
There's an image suffering from degradations {degradations}.  We
will invoke dedicated tools to address these degradations, i.e.,
we will conduct these tasks:  {agenda}.  Now we need to determine
the order of these unordered tasks.  For your information, based
on past trials, we have the following experience:
{experience}
Based on this experience, please give the correct order of the
tasks.  Your output must be a JSON object with two fields:
"thought" and "order", where "order" must be a permutation of
{agenda} in the order you determine.
(Besides, in attempts just now, we found the result is unsatis-
factory if {failed_tries} is conducted first.  Remember not to
arrange {failed_tries} in the first place.)
``` |
| Prompt for DepictQA to compare two images |
| ```
Which of the two images, Image A or Image B, do you consider to
be of better quality?  Answer the question using a single word or
phrase.
``` |

Table 8: Degradation combinations for exploration and testing.

| Group | # Degradations | Seen in exploration? | Degradation combination |
|-------|----------------|----------------------|-------------------------|
| A | 2 | Yes | rain, haze
motion blur, low resolution
low light, noise
defocus blur, JPEG compression artifact
noise, JPEG compression artifact
rain, low resolution
motion blur, low light
defocus blur, haze |
| B | 2 | No | motion blur, JPEG compression artifact
haze, noise
defocus blur, low resolution
rain, low light |
| C | 3 | No | haze, motion blur, low resolution
rain, noise, low resolution
low light, defocus blur, JPEG compression artifact
motion blur, defocus blur, noise |

## B  MORE RESULTS

### B.1  LEARNING FROM EXPLORATION

After collecting the statistics in exploration, we prompt GPT-4 by:

```
We are studying image restoration with multiple degradations.  The
degradation types we are focusing on are:  low-resolution, noise,
motion blur, defocus blur, rain, haze, dark, and jpeg compression
artifact.  We have tools to address these degradations, that is,
we can conduct these tasks:  super-resolution, denoising, motion
deblurring, defocus deblurring, deraining, dehazing, brightening,
and jpeg compression artifact removal.  The problem is, given the
tasks to conduct, we need to determine the order of them.  This
is very complicated because different tasks may have special re-
quirements and side-effects, and the correct order of tasks can
significantly affect the final result.  We have conducted some
trials and collected the following experience:

To address dark+noise in the image, when conducting first denois-
ing and then brightening, the fail rates of addressing ['dark',
'noise'] are ['22%', '43%'] respectively, and the total fail rate
is 32%; when conducting first brightening and then denoising, the
fail rates of addressing ['dark', 'noise'] are ['28%', '42%'] re-
spectively, and the total fail rate is 35%.

To address defocus blur+haze in the image, when conducting first
defocus deblurring and then dehazing, the fail rates of addressing
['defocus blur', 'haze'] are ['0%', '36%'] respectively, and the
total fail rate is 18%; when conducting first dehazing and then
defocus deblurring, the fail rates of addressing ['defocus blur',
'haze'] are ['0%', '40%'] respectively, and the total fail rate is
20%.

To address defocus blur+jpeg compression artifact in the image,
when conducting first jpeg compression artifact removal and then
defocus deblurring, the fail rates of addressing ['defocus blur',
'jpeg compression artifact'] are ['10%', '31%'] respectively, and
the total fail rate is 20%; when conducting first defocus deblur-
ring and then jpeg compression artifact removal, the fail rates
```

Table 9: Single-degradation restoration tools.

| Task | Tools |
| --- | --- |
| Super-resolution | DiffBIR (Lin et al., 2024)
X-Restormer (Chen et al., 2024c)
HAT (Chen et al., 2023b)
SwinIR (Liang et al., 2021) trained under GAN (Goodfellow et al., 2014) paradigm
SwinIR (Liang et al., 2021) trained for optimizing PSNR |
| Denoising | SwinIR (Liang et al., 2021) trained on noise level 15
SwinIR (Liang et al., 2021) trained on noise level 50
MAXIM (Tu et al., 2022)
MPRNet (Zamir et al., 2021)
Restormer (Zamir et al., 2022)
X-Restormer (Chen et al., 2024c) |
| JPEG compression artifact removal | SwinIR (Liang et al., 2021) trained on quality factor 40
FBCNN (Jiang et al., 2021) trained on quality factor 90
FBCNN (Jiang et al., 2021) trained on quality factor 5
FBCNN (Jiang et al., 2021) blind to quality factor |
| Motion deblurring | MAXIM (Tu et al., 2022)
MPRNet (Zamir et al., 2021)
Restormer (Zamir et al., 2022)
X-Restormer (Chen et al., 2024c) |
| Defocus deblurring | DRBNet (Ruan et al., 2022)
IFAN (Lee et al., 2021)
Restormer (Zamir et al., 2022) |
| Deraining | MAXIM (Tu et al., 2022)
MPRNet (Zamir et al., 2021)
Restormer (Zamir et al., 2022)
X-Restormer (Chen et al., 2024c) |
| Dehazing | MAXIM (Tu et al., 2022)
X-Restormer (Chen et al., 2024c)
RIDCP (Wu et al., 2023b)
DehazeFormer (Song et al., 2023) |
| Brightening[1] | Constant shift (Adding a constant 40)
Gamma correction ($\gamma = 2/3$)
CLAHE (Zuiderveld, 1994) |

[1] All tools are operated on the V channel of the HSV color space.

of addressing ['defocus blur', 'jpeg compression artifact'] are ['8%', '48%'] respectively, and the total fail rate is 28%.

To address motion blur+dark in the image, when conducting first motion deblurring and then brightening, the fail rates of addressing ['motion blur', 'dark'] are ['22%', '25%'] respectively, and the total fail rate is 23%; when conducting first brightening and then motion deblurring, the fail rates of addressing ['motion blur', 'dark'] are ['28%', '25%'] respectively, and the total fail rate is 26%.

To address motion blur+low resolution in the image, when conducting first motion deblurring and then super-resolution, the fail rates of addressing ['motion blur', 'low resolution'] are ['23%', '9%'] respectively, and the total fail rate is 16%; when conducting first super-resolution and then motion deblurring, the fail rates of addressing ['motion blur', 'low resolution'] are ['31%', '6%'] respectively, and the total fail rate is 19%.

```
To address noise+jpeg compression artifact in the image, when con-
ducting first denoising and then jpeg compression artifact re-
moval, the fail rates of addressing ['noise', 'jpeg compression
artifact'] are ['38%', '13%'] respectively, and the total fail
rate is 26%; when conducting first jpeg compression artifact re-
moval and then denoising, the fail rates of addressing ['noise',
'jpeg compression artifact'] are ['38%', '14%'] respectively, and
the total fail rate is 26%.

To address rain+haze in the image, when conducting first deraining
and then dehazing, the fail rates of addressing ['rain', 'haze']
are ['5%', '37%'] respectively, and the total fail rate is 21%;
when conducting first dehazing and then deraining, the fail rates
of addressing ['rain', 'haze'] are ['25%', '24%'] respectively,
and the total fail rate is 24%.

To address rain+low resolution in the image, when conducting first
deraining and then super-resolution, the fail rates of addressing
['rain', 'low resolution'] are ['26%', '2%'] respectively, and the
total fail rate is 14%; when conducting first super-resolution and
then deraining, the fail rates of addressing ['rain', 'low resolu-
tion'] are ['63%', '0%'] respectively, and the total fail rate is
32%.
```

The response, which directly serves as experience in Tab. 7, is

```
Based on the collected experience, we can distill the following
knowledge to guide the order of tasks for image restoration:

1. **Dark + Noise**: It is slightly better to denoise before
brightening (32% total fail rate) than to brighten before denois-
ing (35% total fail rate).

2. **Defocus Blur + Haze**: Defocus deblurring should be done
before dehazing, as it results in a lower total fail rate (18% vs.
20%).

3. **Defocus Blur + JPEG Compression Artifact**: Removing JPEG
compression artifacts before defocus deblurring leads to a lower
total fail rate (20% vs. 28%).

4. **Motion Blur + Dark**: Motion deblurring should be done be-
fore brightening, as it results in a lower total fail rate (23%
vs. 26%).

5. **Motion Blur + Low Resolution**: Motion deblurring should be
done before super-resolution, as it results in a lower total fail
rate (16% vs. 19%).

6. **Noise + JPEG Compression Artifact**: The order of denois-
ing and JPEG compression artifact removal does not significantly
affect the total fail rate (both 26%).

7. **Rain + Haze**: Deraining should be done before dehazing, as
it results in a lower total fail rate (21% vs. 24%).

8. **Rain + Low Resolution**: Deraining should be done before
super-resolution, as it results in a significantly lower total
fail rate (14% vs. 32%).

From these observations, we can infer that generally, it is more
effective to address blurring issues (motion blur, defocus blur)
before addressing issues related to image content enhancement
(super-resolution, brightening, dehazing). Additionally, it is
often better to remove artifacts (rain, JPEG compression) before
```

improving the image's content or quality. However, the impact
of the order may vary, and in some cases, such as noise and JPEG
compression artifact, the order does not significantly affect the
outcome.

When dealing with more complex degradations, these guidelines can
serve as a starting point, but it may be necessary to conduct ad-
ditional trials to determine the most effective order of tasks.

## B.2 SCHEDULING RESULT

Tab. 10 comprehensively lists the reasoning and results of scheduling the 16 degradation combinations, and Tab. 11 lists the performance of the resultant plans over each combination. We can see that the scheduling is logical and profitable.

## B.3 SCHEDULING CONSISTENCY

Fig. 13 compares the dispersion of scheduling results with and without experience for each degradation combination. It can be seen that in some extreme cases, scheduling with experience can always yield the same result, while scheduling without experience yields nearly random results. This strongly supports our motivation that providing experience is a must for the sake of consistency.

Besides, as introduced in Sec. 3.3, we find that the scheduling result may be sensitive to the presentation order of subtasks. Therefore, the dispersion of scheduling results may arise from two aspects: one is the intrinsic uncertainty of the LLM, *i.e.*, the response may differ even though the prompt remains the same; another is the bias to presentation order, *i.e.*, the order significantly affects the result. To investigate the bias, the first aspect should be ruled out. We design an intuitive approach: for a metric that measures dispersion, compute the metric values for all presentation orders respectively and then subtract their average from the metric value for the overall results. Formally, denote the metric by $M$ and the distribution of scheduling results by $p$. Also, when the presentation order is fixed, denote the distributions of scheduling results by $p_1, p_2, \cdots, p_N$, respectively, where $N$ is the number of subtask permutations. The difference

$$M(p) - \frac{1}{N} \sum_{i=1}^{N} M(p_i)$$

can reflect the sensitivity to presentation order. For entropy and variation ratio, Fig. 14 compares this difference for scheduling results with and without experience. Notably, for the degradation combination in group A and B, the differences are all zero for scheduling with experience, which means the bias to presentation order is totally eliminated in a sense. These results again justify our design of exploration.

## B.4 COMPARISON WITH OTHER METHODS

Fig. 15 shows more visual results of AgenticIR, compared with all-in-one models and random tool invocation.

Table 10: Scheduling of degradation combinations.

| Group A |
| --- |
| Degradations: rain, haze. 
 Scheduling result: deraining, dehazing. 
 Reasoning: 
 According to the collected experience, specifically point 7, it is more effective to perform deraining before dehazing as it results in a lower total fail rate. Therefore, the correct order of tasks should be to first apply the deraining tool and then the dehazing tool. |

**Table 10 Continued**

Degradations: motion blur, low resolution.
Scheduling result: motion deblurring, SR.
Reasoning:
According to the collected experience, specifically point 5, it
is more effective to address motion blur before dealing with low
resolution.  Therefore, we should perform motion deblurring before
super-resolution to achieve a lower total fail rate.

Degradations: low light, noise.
Scheduling result: denoising, brightening.
Reasoning:
According to the collected experience, it is slightly better to
denoise before brightening when dealing with an image that is both
dark and noisy.  This is supported by the lower total fail rate
observed in past trials for denoising before brightening (32%)
compared to brightening before denoising (35%).

Degradations: defocus blur, JPEG compression artifact.
Scheduling result: JPEG compression artifact removal, defocus deblurring.
Reasoning:
According to the collected experience, it is more effective to ad-
dress image artifacts like JPEG compression before dealing with
blurring issues.  Specifically, for the combination of defocus
blur and JPEG compression artifact, the experience indicates that
removing JPEG compression artifacts before defocus deblurring
leads to a lower total fail rate (20% vs.  28%).  Therefore, the
correct order of tasks should start with JPEG compression artifact
removal followed by defocus deblurring.

Degradations: noise, JPEG compression artifact.
Scheduling result: JPEG compression artifact removal, denoising.
Reasoning:
Since the experience indicates that the order of denoising and
JPEG compression artifact removal does not significantly affect
the total fail rate, we can choose either order for the tasks.
However, to maintain consistency with the general guideline that
suggests removing artifacts before improving the image's content
or quality, we will opt to remove JPEG compression artifacts be-
fore denoising.

Degradations: rain, low resolution.
Scheduling result: deraining, SR.
Reasoning:
According to the collected experience, specifically point 8, it
is more effective to address the artifact caused by rain before
enhancing the image's resolution.  Therefore, we should perform
deraining before super-resolution to achieve a lower total fail
rate.

Degradations: motion blur, low light.
Scheduling result: motion deblurring, brightening.
Reasoning:
According to the collected experience, specifically point 4, it
is more effective to address motion blur before brightening.  This
results in a lower total fail rate.

Degradations: defocus blur, haze.
Scheduling result: defocus deblurring, dehazing.
Reasoning:
According to the collected experience, it is advised to address
defocus blur before haze.  This is because defocus deblurring
should be done before dehazing, as it results in a lower total
fail rate.

**Table 10 Continued**

| Group B |
| --- |

Degradations: motion blur, JPEG compression artifact.
Scheduling result: JPEG compression artifact removal, motion deblurring.
Reasoning:

```
Since the experience suggests that it is often better to remove
artifacts before improving the image's content or quality, and
specifically that removing JPEG compression artifacts before de-
focus deblurring leads to a lower total fail rate, it would be
logical to apply the same principle to motion blur.  Therefore,
we should remove JPEG compression artifacts before addressing the
motion blur.
```

Degradations: haze, noise.
Scheduling result: denoising, dehazing.
Reasoning:

```
While the provided experience does not directly address the com-
bination of haze and noise, we can infer from the guidelines that
issues related to image content enhancement should be addressed
after dealing with artifacts.  Since noise can be considered an
artifact and dehazing is related to enhancing image content by
removing haze, it would be logical to denoise the image before
dehazing it.
```

Degradations: defocus blur, low resolution.
Scheduling result: defocus deblurring, SR.
Reasoning:

```
Following the guidelines, it is generally more effective to
address blurring issues before image content enhancement.
Therefore, we should perform defocus deblurring before super-
resolution.
```

Degradations: rain, low light.
Scheduling result: deraining, brightening.
Reasoning:

```
Since there is no direct experience related to the combination of
rain and dark, we can draw parallels from similar situations.  The
guidelines suggest addressing artifacts like rain before enhancing
the image's content.  This is supported by point 7, which rec-
ommends deraining before dehazing, and point 8, which advises
deraining before super-resolution.  Both dehazing and super-
resolution are content enhancement tasks, similar to brightening.
Therefore, it would be logical to apply the deraining tool before
the brightening tool.
```

| Group C |
| --- |

Degradations: low light, defocus blur, JPEG compression artifact.
Scheduling result: JPEG compression artifact removal, defocus deblurring, brightening.
Reasoning:

```
According to the collected experience, it is generally more effec-
tive to address blurring issues before image content enhancement.
Additionally, removing artifacts like JPEG compression should
be done before addressing blurring issues.  Therefore, the order
should start with JPEG compression artifact removal, followed by
defocus deblurring, and finally brightening the image.
```

**Table 10 Continued**

---

Degradations: haze, motion blur, low resolution.
Scheduling result: motion deblurring, dehazing, SR.
Reasoning:
```
According to the guidelines, it is more effective to address blur-
ring issues before image content enhancement.  Therefore, we
should start with motion deblurring.  There is no direct guide-
line for the order of dehazing and super-resolution in relation
to motion blur, but since dehazing is an image content enhancement
and super-resolution is also an enhancement but more towards image
quality, we should perform dehazing before super-resolution.  This
aligns with the general principle of removing artifacts before
improving the image's content or quality.
```

---

Degradations: motion blur, defocus blur, noise.
Scheduling result: motion deblurring, defocus deblurring, denoising.
Reasoning:
```
Since both motion blur and defocus blur are types of blurring is-
sues, and based on the experience that blurring issues should be
addressed before image content enhancement, which in this case is
denoising, we should first deal with the blurring.  Between motion
blur and defocus blur, there is no direct experience suggesting
which should be addressed first.  However, considering that mo-
tion blur might affect the entire image while defocus blur could
be more localized, it might be more effective to correct the mo-
tion blur first to ensure that any subsequent defocus deblurring
operates on a more stable image base.  Therefore, the order should
start with motion deblurring, followed by defocus deblurring, and
finally denoising.
```

---

Degradations: rain, noise, low resolution.
Scheduling result: deraining, denoising, SR.
Reasoning:
```
According to the guidelines, it is generally more effective to re-
move artifacts like rain before addressing image content enhance-
ment like super-resolution.  Additionally, there is no specific
mention of the interaction between noise and low resolution, but
since noise is an artifact, it would be logical to address it be-
fore enhancing the image content.  Therefore, the order should
start with deraining, followed by denoising, and finally, super-
resolution.
```

---

## C   ALTERNATIVE LLM

We check whether our framework can also achieve competitive performance with other LLMs. Tab. 12 lists the performance outcome of replacing GPT-4 with the popular open-source Llama 3.1 405B (Llama Team, AI @ Meta, 2024). It can be seen that the shift from GPT-4 to Llama does not influence the performance severely, and AgenticIR with Llama surpasses random invocation by a large margin too. This is because provided with the experience, the scheduling problem is easily enough to be handled by different LLMs, and Llama can give results similar to GPT-4. In fact, the performance difference is more likely to be caused by randomness of tool invocation.

## D   DISCUSSION ON EXPLORATION-EXPLOITATION TRADEOFF

A primary problem in decision-making is the tradeoff between exploration and exploitation (Berger-Tal et al., 2014). AgenticIR, as a heuristic search, deals with this by greedily exploiting acceptable directions and pruning those seemingly unpromising directions in reflection. To some extent, this behavior favors exploitation and thus may suffer from early stopping, resulting in suboptimal results. However, such preference is configurable. That is, we can adjust the acceptance threshold in reflection to suppress exploitation so as to force exploring more. We conduct an experiment that lets our agent only accept tool outputs with very low severity of degradations, denoted as AgenticIR$^*$.

Table 11: Detailed comparison between acting as the agent's plan and the opposite. "Not as planned" means randomly shuffling the plan (guaranteed to be different).

| Degradations | As planned | PSNR | SSIM | LPIPS↓ | MANIQA | CLIP-IQA | MUSIQ |
|---|---|---|---|---|---|---|---|
| rain, | ✓ | **19.00** | **0.8075** | **0.1655** | **0.4309** | **0.6015** | **68.25** |
| haze | ✗ | 16.35 | 0.7324 | 0.2250 | 0.4296 | 0.5900 | 68.03 |
| motion blur, | ✓ | **20.57** | **0.5532** | **0.2575** | 0.4192 | **0.6575** | 66.38 |
| low resolution | ✗ | 20.28 | 0.5427 | 0.2637 | **0.4374** | 0.6568 | **67.19** |
| low light, | ✓ | **20.13** | **0.7454** | 0.3042 | 0.3516 | 0.4679 | 59.88 |
| noise | ✗ | 18.97 | 0.7175 | **0.2997** | **0.3962** | **0.5022** | **61.63** |
| defocus blur, | ✓ | **24.34** | **0.6846** | **0.3885** | **0.2163** | **0.2859** | **45.83** |
| jpeg compression artifact | ✗ | 23.78 | 0.6580 | 0.4614 | 0.1938 | 0.2743 | 41.53 |
| noise, | ✓ | **28.08** | **0.8129** | **0.2491** | **0.3537** | **0.5343** | **60.03** |
| jpeg compression artifact | ✗ | 27.50 | 0.7997 | 0.2714 | 0.3484 | 0.5182 | 58.95 |
| rain, | ✓ | 20.42 | **0.6005** | **0.3045** | **0.4216** | **0.6710** | **66.42** |
| low resolution | ✗ | **21.59** | 0.5796 | 0.3801 | 0.3449 | 0.5292 | 64.35 |
| motion blur, | ✓ | **17.06** | 0.5350 | **0.2849** | 0.2785 | **0.4151** | 60.03 |
| low light | ✗ | 17.05 | **0.5442** | 0.2903 | **0.2826** | 0.3935 | **60.41** |
| defocus blur, | ✓ | 19.52 | 0.7300 | **0.2484** | **0.3033** | **0.4399** | **59.33** |
| haze | ✗ | **20.81** | **0.7477** | 0.2563 | 0.2751 | 0.3908 | 56.73 |
| motion blur, | ✓ | **22.12** | **0.6638** | **0.3385** | **0.2355** | **0.3784** | **51.24** |
| jpeg compression artifact | ✗ | 21.61 | 0.6464 | 0.3831 | 0.1943 | 0.3215 | 45.64 |
| haze, | ✓ | **20.20** | **0.7395** | **0.3130** | 0.3480 | 0.4645 | **59.81** |
| noise | ✗ | 17.45 | 0.6654 | 0.3523 | **0.3785** | **0.5519** | 58.38 |
| defocus blur, | ✓ | 22.42 | 0.6118 | **0.2475** | 0.4283 | 0.6681 | **67.92** |
| low resolution | ✗ | **23.22** | **0.6232** | 0.2491 | **0.4432** | **0.6777** | 67.74 |
| rain, | ✓ | **19.82** | **0.8202** | **0.1743** | 0.4234 | **0.5991** | 68.70 |
| low light | ✗ | 18.99 | 0.7892 | 0.2058 | **0.4334** | 0.5517 | **68.86** |
| haze, motion blur, | ✓ | **17.20** | **0.4923** | **0.2951** | **0.4193** | 0.6640 | **65.97** |
| low resolution | ✗ | 15.98 | 0.4730 | 0.3215 | 0.4084 | **0.6698** | 65.32 |
| rain, noise, | ✓ | 20.73 | 0.5602 | **0.3785** | **0.4100** | **0.6736** | **62.34** |
| low resolution | ✗ | **21.14** | **0.5868** | 0.3908 | 0.3683 | 0.5610 | 59.89 |
| low light, defocus blur, | ✓ | **18.57** | **0.6194** | **0.4223** | **0.1978** | **0.2726** | **47.04** |
| jpeg compression artifact | ✗ | 18.40 | 0.6108 | 0.4251 | 0.1951 | 0.2685 | 46.45 |
| motion blur, defocus blur, | ✓ | **18.64** | **0.4690** | **0.5997** | 0.2202 | 0.3401 | 28.96 |
| noise | ✗ | 18.42 | 0.4402 | 0.6006 | **0.2514** | **0.3883** | **33.63** |

The results are shown in Tab. 13, compared with the original setting and random tool invocation. AgenticIR* does obtain slightly better results in most metrics, but also consumes much more time as shown in Tab. 14. Therefore, we believe it is fair to say the current setting of AgenticIR strikes a balance between performance and efficiency.

## E  LIMITATION

Our primary goal is to realize an intelligent agent for IR tool ensemble, but this paper only considers single-degradation restoration tools, limiting the application to the so-called complex-degradation restoration problem. It is worth exploring whether our paradigm can integrate more general and heterogeneous tools, which will require higher flexibility and decision-making capabilities. A generally capable agent is supposed to surmount these challenges.

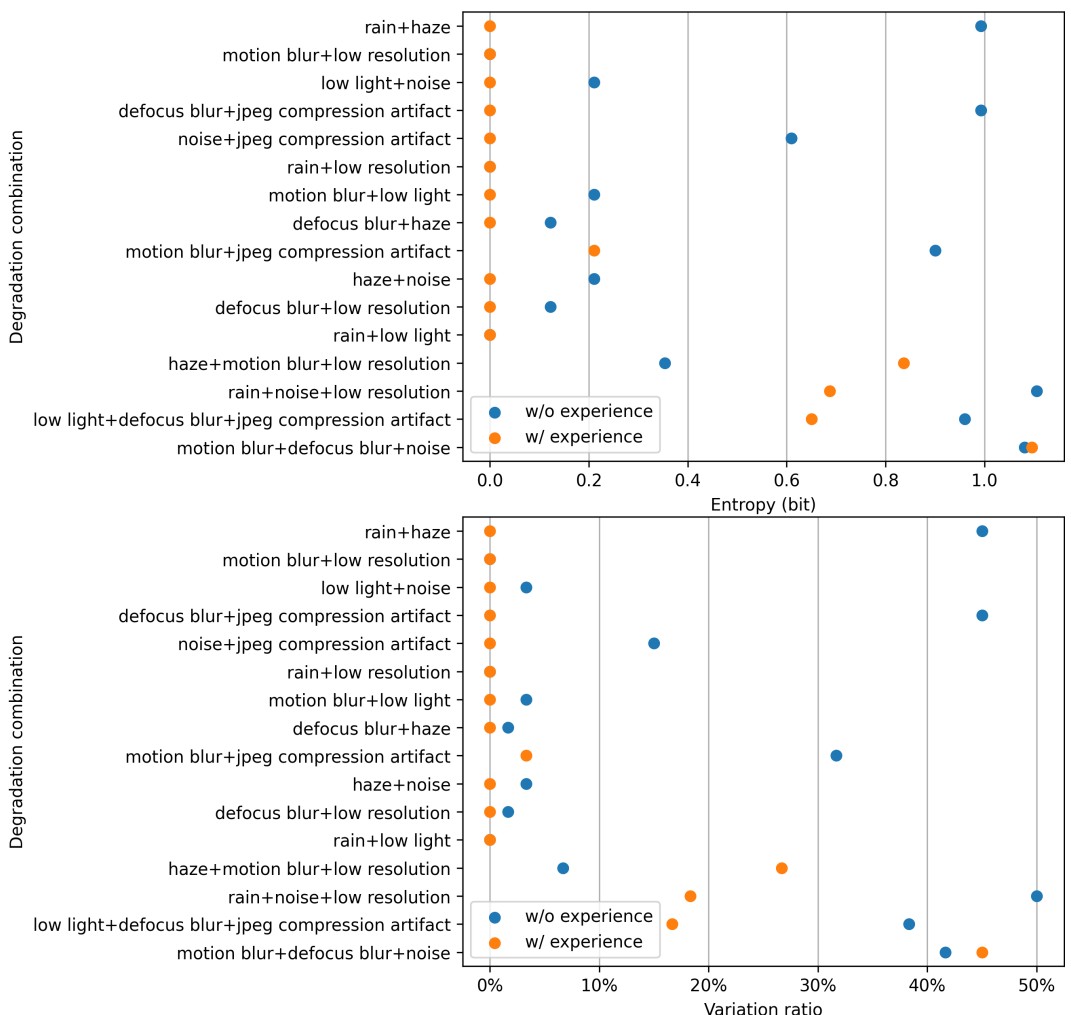

Figure 13: Detailed comparison between dispersion of scheduling results with and without experience.

Table 12: Quantitative comparison between our framework with different LLMs and random tool invocation. "AgenticIR" has the same setting as in the main text, while "AgenticIR (Llama)" replaces GPT-4 with Llama for (re)scheduling.

| Degradations | Method | PSNR | SSIM | LPIPS↓ | MANIQA | CLIP-IQA | MUSIQ |
|---|---|---|---|---|---|---|---|
| Group A | AgenticIR (Llama) | **21.06** | **0.6834** | **0.3084** | **0.3123** | **0.4516** | **57.61** |
| | AgenticIR | 21.04 | 0.6818 | 0.3148 | 0.3071 | 0.4474 | 56.88 |
| | Random | 20.90 | 0.6642 | 0.3368 | 0.2963 | 0.4394 | 55.30 |
| Group B | AgenticIR (Llama) | **20.79** | **0.7019** | **0.3062** | 0.3174 | 0.4648 | 57.47 |
| | AgenticIR | 20.55 | 0.7009 | 0.3072 | **0.3204** | **0.4648** | **57.57** |
| | Random | 20.06 | 0.6766 | 0.3351 | 0.3120 | 0.4514 | 56.15 |
| Group C | AgenticIR (Llama) | 18.80 | **0.5480** | 0.4562 | 0.2675 | 0.3859 | 48.13 |
| | AgenticIR | 18.82 | 0.5474 | **0.4493** | **0.2698** | **0.3948** | **48.68** |
| | Random | **18.87** | 0.5456 | 0.4796 | 0.2354 | 0.3543 | 44.61 |

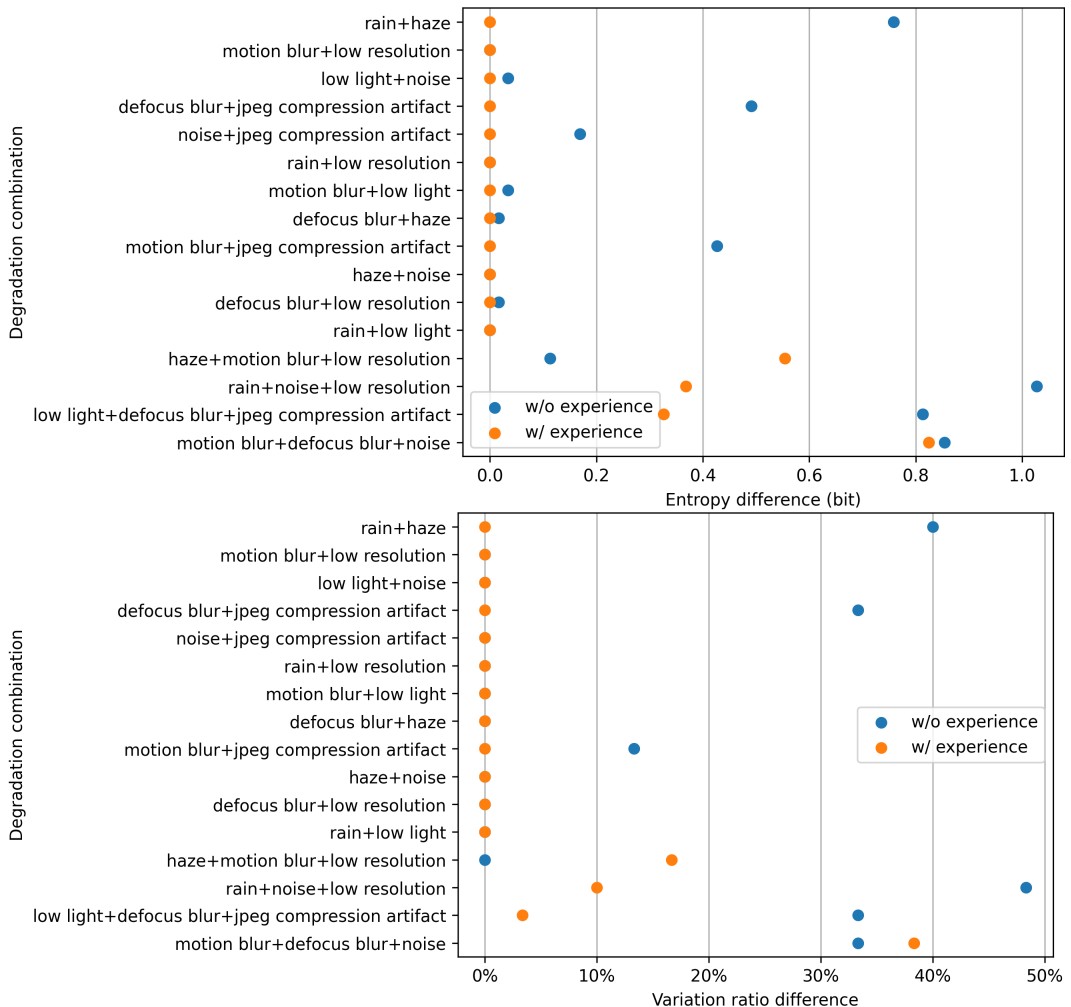

Figure 14: Detailed comparison between sensitivity to presentation order when scheduling with and without experience.

Table 13: Quantitative comparison between restoration performance of AgenticIR with different acceptance thresholds in reflection, and thus different preferences for exploration.

| Degradations | Method | PSNR | SSIM | LPIPS↓ | MANIQA | CLIP-IQA | MUSIQ |
|---|---|---|---|---|---|---|---|
| Group A | AgenticIR | **21.04** | **0.6818** | **0.3148** | 0.3071 | 0.4474 | 56.88 |
| | AgenticIR* | 20.95 | 0.6790 | 0.3169 | **0.3082** | **0.4475** | **56.99** |
| Group B | AgenticIR | 20.55 | 0.7009 | 0.3072 | 0.3204 | 0.4648 | 57.57 |
| | AgenticIR* | **20.72** | **0.7014** | **0.3017** | **0.3223** | **0.4684** | **57.78** |
| Group C | AgenticIR | 18.82 | **0.5474** | **0.4493** | **0.2698** | **0.3948** | **48.68** |
| | AgenticIR* | **18.90** | 0.5459 | 0.4582 | 0.2667 | 0.3902 | 47.71 |

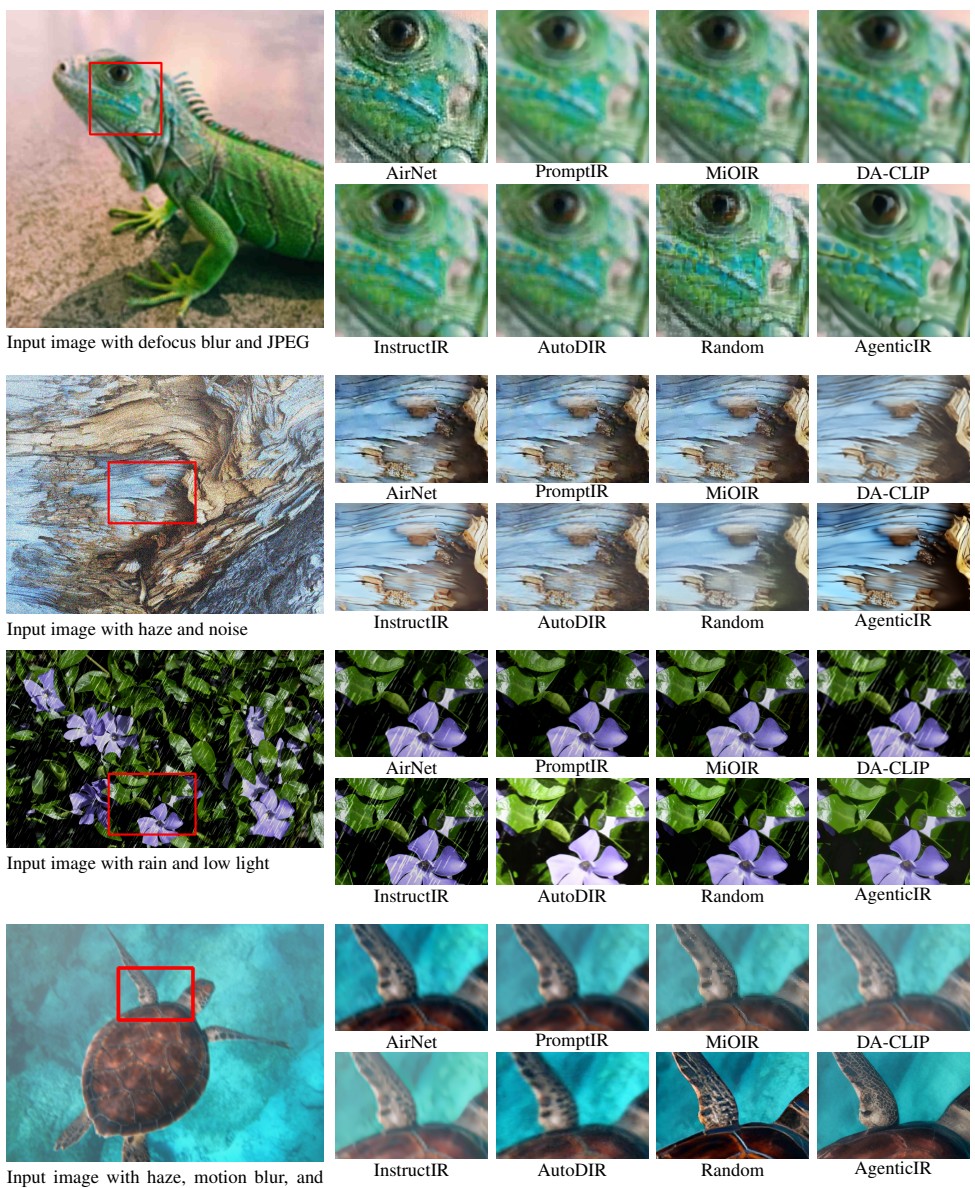

Figure 15: Qualitative comparison with other methods.

Table 14: Quantitative comparison between cost of AgenticIR with different acceptance thresholds in reflection.

| Degradations | Method | Wall clock time (second) | # Tool invocations |
|---|---|---|---|
| Group A | AgenticIR | 48 | 3.37 |
| | AgenticIR* | 137 | 8.07 |
| Group B | AgenticIR | 54 | 3.63 |
| | AgenticIR* | 117 | 7.01 |
| Group C | AgenticIR | 78 | 4.77 |
| | AgenticIR* | 174 | 10.50 |

