# OpenReview forum: "An Intelligent Agentic System for Complex Image Restoration Problems"
_ICLR.cc/2025/Conference — ICLR 2025 Poster_

### Official Review · Reviewer_hoAk · 2024-10-27

**Soundness:** 3
**Presentation:** 3
**Contribution:** 3
**Rating:** 8
**Confidence:** 3

**Summary:**

This paper introduces AgenticIR, an intelligent system designed to handle complex image restoration tasks by emulating human-like problem-solving methods. The system operates through five stages: Perception, Scheduling, Execution, Reflection, and Rescheduling. AgenticIR leverages LLM and VLM, using their text generation capabilities to operate a set of IR tools dynamically. It relies on VLMs for image quality assessment and LLMs for step-by-step reasoning, enhancing its adaptability to various IR challenges.

It also incorporates a self-exploration mechanism that generates referenceable summaries from past restoration attempts, which improves its decision-making. Experimental results show AgenticIR’s effectiveness in complex restoration scenarios, highlighting its potential for real-world automated image processing and broader AI applications in visual processing.

**Strengths:**

1. Human-Centric Design: AgenticIR provides an image restoration approach that mirrors human actions, incorporating processes like reflection and iterative rescheduling into its pipeline. This design enhances action interpretability and facilitates meaningful human interaction with the system.
2. Clear and Concise Expression: The paper presents complex ideas with clarity, accompanied by detailed images and diagrams that enhance comprehension and support the technical explanations.
3. Comprehensive Experiments and Ablation Studies: Thorough experimental evaluations are provided, with well-structured ablation studies for each module. This approach effectively validates the system's design and performance.
4. Illustrative Pipeline Examples: The pipeline is illustrated with specific cases, offering a clear understanding of how each component functions within real-world scenarios.

**Weaknesses:**

1. Limitations Compared to Optimal Solutions: AgenticIR tends to encounter an early stopping issue, where it may settle on a satisfactory solution prematurely, halting further exploration and potentially missing the optimal outcome. Addressing this limitation is important, and it might be beneficial to add an additional row to Table 4 to reflect this aspect.
2. Insufficient Reporting on Iteration Count and Processing Time: Although the paper emphasizes the role of experiential information and provides illustrative examples, it lacks concrete data on the actual reduction in iterations or time consumption. Including specific metrics on these improvements would strengthen the evaluation of AgenticIR’s efficiency and practical advantages.

**Questions:**

In addition to the weaknesses mentioned, I have two further questions:

1. Why Not Use VLMs Exclusively Throughout the Pipeline? Given VLMs' strong capabilities in image quality assessment and reasoning, could a VLM-only approach be more efficient or effective for the entire pipeline?
2. Would Online Updates to the Reference Data Benefit the Pipeline? Could implementing real-time updates to the experiential knowledge base further enhance the pipeline’s adaptability and performance?

I hope the authors can make up for the weaknesses mentioned and address these questions.

---

> ### Author Response · Authors · 2024-11-20
> **Response to Reviewer hoAk (1/2)**
>
> Thank you for recognizing the human-centric design of AgenticIR, particularly its reflection and iterative rescheduling processes that enhance interpretability and interaction. We also deeply appreciate your acknowledgment of our clear presentation, the novelty of our approach, and the rigor of our experiments.
>
>
> `1.` *AgenticIR tends to encounter an early stopping issue, where it may settle on a satisfactory solution prematurely, halting further exploration and potentially missing the optimal outcome.*
>
> `A`: Thanks for the question. Due to the complexity of image restoration and unpredictability of tools, the only way to guarantee optimal solution is exhaustive search, which is impractical (even if there are only two degradations and three tools for each, the required number of tool invocation will be 24). Hence any method is a trade-off between exploration and exploitation. AgentlicIR, as a heuristic search, deals with this by greedily exploiting acceptable directions and pruning those seemingly unpromising directions in reflection. This behavior does tend to exploitation and thus suffer from early stopping. In fact this preference is configurable. That is, we can adjust the acceptance threshold in reflection to suppress exploitation so as to force exploring more. We conduct an experiment that let AgenticIR only accepts tool outputs with very low severity of degradations, denoted as AgenticIR\*. The results are shown in the table below, compared with AgenticIR and random tool invocation. AgenticIR\* does obtain better results in most metrics, but also consumes much more time as shown below. Therefore, we believe it is fair to say the current setting of AgenticIR strikes a balance between performance and efficiency.
>
> | Degradations | Method | PSNR |  SSIM |  LPIPS |  MANIQA |  CLIPIQA |  MUSIQ |
> |:---|:---|:---:| :---:| :---:| :---:| :---:| :---:|
> | Group A | Random | *20.9* | 0.6642 | 0.3368 | 0.2963 | 0.4394 | 55.3 |
> | Group A | AgenticIR | **21.04** | **0.6818** | *0.3148* | *0.3071* | *0.4474* | *56.88* |
> | Group A | AgenticIR* | 20.38 | *0.6665* | **0.3063** | **0.3354** | **0.4802** | **60.44** |
> | Group B | Random | 20.06 | 0.6766 | 0.3351 | 0.312 | 0.4514 | 56.15 |
> | Group B | AgenticIR | *20.55* | **0.7009** | *0.3072* | *0.3204* | *0.4648* | *57.57* |
> | Group B | AgenticIR*| **20.78** | *0.6991* | **0.2862** | **0.3415** | **0.4926** | **60.55** |
> | Group C | Random | *18.87* | 0.5456 | 0.4796 | 0.2354 | 0.3543 | 44.61 |
> | Group C | AgenticIR | 18.82 | *0.5474* | *0.4493* | *0.2698* | *0.3948* | *48.68* |
> | Group C | AgenticIR*| **19.08** | **0.5516** | **0.4302** | **0.2892** | **0.4369** | **51.86** |
>
>
> |   | Degradations | Method     | Wall clock time (s) | #Tool invocations |   |   |   |   |   |
> |---|--------------|------------|---------------------|-------------------|---|---|---|---|---|
> |   | Group A      | AgenticIR  | 48                  | 3.37              |   |   |   |   |   |
> |   | Group A      | AgenticIR* | 137                 | 8.07              |   |   |   |   |   |
> |   | Group B      | AgenticIR  | 54                  | 3.63              |   |   |   |   |   |
> |   | Group B      | AgenticIR* | 117                 | 7.01              |   |   |   |   |   |
> |   | Group C      | AgenticIR  | 78                  | 4.77              |   |   |   |   |   |
> |   | Group C      | AgenticIR* | 174                 | 10.50             |   |   |   |   |   |
>
> `2.` *Although the paper emphasizes the role of experiential information and provides illustrative examples, it lacks concrete data on the actual reduction in iterations or time consumption.*
>
> `A`: Thanks for the question. Please see the public comment.

---

> ### Author Response · Authors · 2024-11-20
> **Response to Reviewer hoAk (2/2)**
>
> `3.` *Why Not Use VLMs Exclusively Throughout the Pipeline?*
>
> `A`: Thank you for your question. We understand that you might think providing images could be helpful if the scheduling is performed by a powerful visual language model (VLM) like GPT-4V. However, we have not adopted this approach for several reasons:
>
> We are skeptical about the effectiveness of using images with current VLMs during scheduling. In fact, we have found that even the most powerful VLM to date—GPT-4V—performs suboptimally in the image quality assessments required by our framework. We tested GPT-4V’s degraded recognition capabilities (with experimental settings identical to the fine-tuned DepictQA in our paper). The results are shown in the table below, far from satisfactory (see Table 2 in the paper for details). This indicates that achieving expertise in low-level visual aspects is not easy.
>
> Furthermore, most current VLM methods are obtained by fine-tuning large language models (LLMs). Due to the limitations of fine-tuning data, the reasoning ability and knowledge breadth of VLMs are significantly inferior to LLMs. Therefore, there is currently no VLM suitable for our framework that possesses both strong low-level visual capabilities and general reasoning abilities. We look forward to the development of VLMs as powerful foundational models in the future.
>
> Based on the above considerations regarding design methodology and practical effectiveness, we choose to use LLMs for reasoning and VLMs for perception separately, rather than relying solely on VLMs.
>
> |Degradation| Precision | Recall | F1 score|
> |:----:|:----:|:----:|:----:|
> |Noise| 0.40 | 0.97 | 0.57 |
> | Motion blur | 0.40 | 0.61 | 0.48 |
> | Defocus blur | 0.56 | 0.87 | 0.68 |
> | JPEG artifact | 0.22 | 0.56 | 0.31 |
> | Rain | 1.00 | 0.79 | 0.88 |
> | Haze | 0.71 | 0.17 | 0.28 |
> | Low light | 0.53 | 0.34 | 0.42 |
>
> `4.` *Would Online Updates to the Reference Data Benefit the Pipeline? Could implementing real-time updates to the experiential knowledge base further enhance the pipeline’s adaptability and performance?*
>
> `A`: Thank you for the reviewer's question; it's a very, very good one. Discovering and learning new knowledge in real time during practical processes, and even possessing a certain degree of creativity, is a higher-level goal in agent research. Achieving this requires us to make more breakthroughs in many core technologies. For example, in the cognitive architecture of learning from cases, we need to abstract the steps humans use to learn from past cases and construct automated mechanisms to implement these steps in agent robots. Additionally, we need to establish reasonable knowledge representation methods to express case-based knowledge (the knowledge in this paper is rule-based). We also need methods to retrieve information from a large number of cases. There is a vast space for us to conduct such research, and I believe these ideas can be realized in the future.
>
> Returning to this paper, it represents very early exploratory work on agents in image processing. We built the research platform from scratch, discussed research methods, and demonstrated preliminary results and the potential of agent-based image processing systems. The issues mentioned above are currently beyond the scope of this paper, but we are extremely interested in them and look forward to gradually addressing them in our future work.

---

> > ### Comment · Area_Chair_sZSc · 2024-11-24
> > **Discussion Period Ending Soon**
> >
> > Dear Reviewer,
> >
> > The discussion period will end soon. Please take a look at the author's comments and begin a discussion.
> >
> > Thanks, Your AC

---

> > ### Comment · Reviewer_hoAk · 2024-11-25
> >
> > Thank you for providing detailed responses to my comments and addressing my concerns. I appreciate the effort you put into clarifying these points and improving the manuscript. I hope my feedback has been helpful in refining your work.

---

> > > ### Author Response · Authors · 2024-11-25
> > > **Thanks to Reviewer hoAk**
> > >
> > > Thank you for recognizing our work.
> > >
> > > Your comments inspire us to deepen our understanding of the exploration-exploitation tradeoff and strengthen our confidence in pursuing lifelong learning for agents.

---

### Official Review · Reviewer_sgx9 · 2024-11-02

**Soundness:** 3
**Presentation:** 3
**Contribution:** 3
**Rating:** 6
**Confidence:** 4

**Summary:**

To address the complex image restoration(IR) problems in real-world scenarios, the authors propose AgenticIR, which is an agent-based system that comprises five stages: perception, scheduling, execution, reflection, and rescheduling. The system leverages a Vision-Language Model (VLM) and a Large Language Model (LLM) to reason and connect these five stages, utilizing an IR toolbox during the execution phase to perform actual image reconstruction.

The three main elements in this process are the LLM, VLM, and the IR toolbox. The VLM primarily analyzes the image quality and summarizes the degradation issues, fine-tuning its capabilities based on existing work. The LLM is responsible for planning and strategizing based on the VLM's results, utilizing GPT-4 and employing a self-exploration method to gain experience with IR problems. The IR toolbox consists of a combination of 3-6 existing models tailored to each type of image degradation.

**Strengths:**

-	This paper is well-written and easy to follow.
-	The experimental setup is comprehensive, with sufficient ablation and comparative experiments demonstrating the effectiveness of their proposed methods
-	The discovery that execution order is key to restoring complex degraded images is compelling.

**Weaknesses:**

-	The main concern is that this work resembles several widely-used frameworks (e.g., large language models (LLMs), vision-language models (VLMs) and image restoration models), giving it a predominantly engineering-focused approach.
-	Additionally, this work appears complex, so providing statistics on the time and complexity involved in a single inference would enhance clarity.
-	Given that LLMs and VLMs often struggle with the issue of 'hallucination,' does this work encounter a similar challenge? If so, how does it address this problem?
-	What are the limitations of the proposed framework?
-	Regarding the point that 'execution order is crucial', are the documents (knowledge) remain consistent during inference across different test samples?

**Questions:**

Please see the Weaknesses.

---

> ### Author Response · Authors · 2024-11-21
> **Response to Reviewer sgx9 (1/2)**
>
> We sincerely thank the reviewer for their feedback and kind recognition of our paper’s clarity, comprehensive experiments, and key contributions.
>
> `1.` *The main concern is that this work resembles several widely-used frameworks, giving it a predominantly engineering-focused approach.*
>
> `A`:
> We respectfully disagree with classifying our work as “engineering-focused.” First, our research is not a simple combination/resemble of existing methods. We propose a novel methodology focusing on the integration and construction of image restoration agents (AI Agents). This methodology represents a significant academic contribution. We thoroughly explain why such an agentic approach is needed and how it can be designed and constructed. Based on this, we built a research platform and, for the first time, demonstrated through experiments that an agentic approach can exhibit a considerable level of intelligence in low-level vision tasks such as image restoration. This is a major breakthrough that cannot be achieved by a single image restoration model.
>
> Regarding the combination of multiple models, we have extensively justified the necessity and advancement of this paradigm. Currently, AI Agent research is at the forefront of attention, with numerous studies exploring how to operate LLMs and other AI models used as tools more intelligently. The core of these studies lies in designing cognitive architectures to enable collaboration among multiple functional models, thereby exhibiting higher levels of intelligence. This is the focus of our work and a key problem for many subsequent studies to address. These cutting-edge efforts have garnered significant attention from both academia and industry and are far from being “engineering-focused.” We recommend that the reviewer carefully review the related work section, which we believe will address the reviewer's concerns.
>
> We sincerely hope the reviewer can reevaluate this perspective and look forward to further discussions with you.
>
> `2.` *Additionally, this work appears complex, so providing statistics on the time and complexity involved in a single inference would enhance clarity.*
>
> `A`:
> Thank you for the reviewer’s questions. We encourage the reviewer to refer to the public response at: https://openreview.net/forum?id=3RLxccFPHz&noteId=XtMKDGs0G7, where we have provided detailed explanations regarding efficiency, cost, and fairness concerns.
>
> `3.` *Given that LLMs and VLMs often struggle with the issue of 'hallucination,' does this work encounter a similar challenge?*
>
> `A`:
> Thank you for the insightful comments. We highly agree with reviewer’s perspective. LLMs and VLMs indeed often face the issue of “hallucination,” which is one of the core motivations behind our proposed **self-exploration and summarization** method. The reliability issue in LLM-based scheduling can be seen as a form of hallucination (i.e., factual inconsistency or fabrication). This issue is extensively studied in Lines 302–323, Lines 408–418, and Appendix B.3 of our paper. Our experiments revealed that GPT-4 occasionally provides random answers when determining operation sequences in zero-shot settings, suggesting that its responses might be based on irrelevant factors.
>
> To address this, we designed the **self-exploration and experience summarization** mechanism, which introduces clear references to enhance the reliability of scheduling. This mechanism enables the LLM to make decisions grounded in concrete foundations rather than relying solely on speculative reasoning. For VLMs, during the training of DrpictQA, we utilized carefully designed fine-tuning data to align the model outputs with human perception, thereby mitigating hallucination to a certain extent. While addressing hallucination remains a significant challenge in both LLM and VLM research, our approach has considered this issue and proposed feasible preliminary solutions.

---

> ### Author Response · Authors · 2024-11-21
> **Response to Reviewer sgx9 (2/2)**
>
> `4.` *What are the limitations of the proposed framework?*
>
> Thanks for the question. We discuss this issue in Appendix E: our primary goal is to develop an intelligent agent for integrating tools in image restoration. However, the current framework only considers single-degradation restoration tools. In real-world scenarios, degradations are often much more complex than a combination of a few well-defined degradations, requiring more general and heterogeneous tools.
>
> For instance, to handle more intricate degradations, we may need to incorporate diffusion models with strong generative capabilities and carefully fine-tune their complex parameters. Additionally, we might need to leverage various tools in Photoshop, much like a professional retoucher. Whether our framework is sufficiently flexible to integrate these diverse tools remains an open question. This challenge also imposes higher demands on the agent’s perception and decision-making capabilities.
>
> `5.` *Regarding the point that 'execution order is crucial', are the documents (knowledge) remain consistent during inference across different test samples?*
>
> `A`:
> Thanks for the question. Our response is: Yes, self-exploration and experience summarization are performed only once beforehand. Afterward, all tests are conducted independently, utilizing the same “knowledge.” However, as reviewer hoAk pointed out, it is also feasible to treat the tests as a unified process, where the agent incrementally updates its knowledge online based on the restoration results. For a more detailed discussion, please refer to our response to reviewer hoAk’s Question 4 in Section 2.
>
> In summary, we found that achieving efficient online updates requires more advanced techniques, such as update strategy design, retrieval-augmented generation, knowledge representation, prompt engineering, and chain-of-thought reasoning. These directions offer promising opportunities for future research.

---

> > ### Comment · Area_Chair_sZSc · 2024-11-24
> > **Discussion Period Ending Soon**
> >
> > Dear Reviewer,
> >
> > The discussion period will end soon. Please take a look at the author's comments and begin a discussion.
> >
> > Thanks, Your AC

---

> > ### Comment · Reviewer_sgx9 · 2024-11-27
> > **Response to Authors**
> >
> > Thank the authors for the detailed response. Most of my concerns are solved. And I decide to maintain my original rating.

---

### Official Review · Reviewer_wMVY · 2024-11-04

**Soundness:** 2
**Presentation:** 3
**Contribution:** 1
**Rating:** 5
**Confidence:** 4

**Summary:**

The proposed AgenticIR system addresses the inherent complexity of real-world image restoration (IR) tasks by emulating a human-like, multi-stage processing workflow. The system operates through distinct phases: Perception, Scheduling, Execution, Reflection, and Rescheduling. It integrates Large Language Models (LLMs) and Vision Language Models (VLMs) into a collaborative framework, allowing text-based interactions to direct the application of specialized IR models. This agentic approach builds on existing multi-modal IR frameworks by dynamically adapting its restoration strategy, actively reassessing and adjusting to handle various complex degradations.

**Strengths:**

1. The proposed system offers a comprehensive approach to image restoration, addressing a broad range of degradation types through a structured, adaptable methodology.
2. It incorporates human-interaction-inspired insights into the image restoration process, potentially enhancing adaptability and effectiveness in handling complex restoration tasks.

**Weaknesses:**

1. **Comparison fairness**: The comparative experiments appear to lack fairness, as the baselines (e.g., InstructIR) are designed as unified IR models trained to handle multiple degradation types in a single framework. In contrast, AgenticIR leverages specialized off-the-shelf models for each type of degradation. Therefore, it would be more appropriate to compare AgenticIR to the state-of-the-art models for each specific degradation task rather than to unified restoration models.

2. **Efficiency Concerns**: While the system is comprehensive, its workflow is notably complex and lengthy. Compared to regular image restoration models, how efficient is AgenticIR in processing images? This is a critical aspect for real-world applications of image restoration and should be addressed with precise comparative evaluations.

3. **Toolbox Ablation Study**: Lines 159-160 state, "For each degradation, we collect three to six advanced models to build the ‘toolbox’ that the intelligent agent can use." There is no ablation study analyzing the impact of selecting these advanced models on the system’s effectiveness. Understanding the influence of each selected model in the toolbox could provide valuable insights.

4. **GPT-4 Usage and Reproducibility**: AgenticIR uses GPT-4, but GPT-4 lacks a fixed version, which raises concerns about the reproducibility of experimental results. Additionally, there is no ablation study on the effects of using alternative LLMs, particularly open-source options, on performance outcomes.

**Questions:**

NA

---

> ### Author Response · Authors · 2024-11-21
> **Response to Reviewer wMVY (1/2)**
>
> We thank the reviewer for the feedback and for acknowledging our structured, adaptable methodology and human-interaction-inspired approach.
>
> `1.` *Comparison fairness*
>
> `A`: Thank you for the reviewer’s questions. First, we encourage the reviewer to refer to the public response at: https://openreview.net/forum?id=3RLxccFPHz&noteId=XtMKDGs0G7, where we have provided detailed explanations regarding efficiency, cost, and fairness concerns.
>
> We would also like to provide additional responses to the reviewer’s suggestion that “*it would be more appropriate to compare AgenticIR to the state-of-the-art models for each specific degradation.*” While we appreciate this perspective, we respectfully hold a different view.
>
> First, it is inherently challenging to compare our method with such state-of-the-art models. The primary goal of AgenticIR is to intelligently address complex, ill-defined image restoration problems where existing methods often fail to provide satisfactory results. Many of the degradations we evaluate cannot be easily paired with a state-of-the-art model, as their use outside their originally intended scope often leads to significant limitations—an issue well-documented in studies exploring the generalization performance of image restoration methods.
>
> Second, our approach is specifically designed to intelligently integrate various image restoration models. If there are state-of-the-art models that excel in specific scenarios, they can be seamlessly incorporated into our toolbox. For cases where these models perform well, our system will leverage them to achieve superior results. For cases they cannot handle, our approach will dynamically adopt alternative strategies. At all times, our method can combine the strengths of available tools, delivering results that are at least comparable to, if not better than, any single method used alone.
>
> `2.` *Efficiency Concerns*
>
> `A`: Thank you for the reviewer’s questions. We encourage the reviewer to refer to the public response at: https://openreview.net/forum?id=3RLxccFPHz&noteId=XtMKDGs0G7, where we have provided detailed explanations regarding efficiency, cost, and fairness concerns.

---

> ### Author Response · Authors · 2024-11-21
> **Response to Reviewer wMVY (2/2)**
>
> `3.` *Toolbox Ablation Study*
>
> `A`: Thank you for the reviewer’s questions. We would like to emphasize that the selection of models is not an intrinsic part of our proposed method. The models we chose represent a broad range of options for specific types of degradation, rather than being selected for their potential to enhance performance. Adding more models to the toolbox expands the applicability of our approach and increases the likelihood of achieving better results. Ideally, a fully capable agent should have access to all currently available models. For the sake of convenience in our research, we prepared a representative subset of models as the toolbox.
>
> As discussed under the “Comparison Fairness” section, we can continuously incorporate better models into the toolbox to achieve improved results across more images. However, our criteria for model selection are not limited to performance alone. Different models exhibit different behaviors—some may have lower numerical quality but offer unique effects or generalization capabilities for specific types of images. We aim to include a diverse range of models in the toolbox to better address the varied challenges found in real-world scenarios.
>
> `4.` *GPT-4 Usage and Reproducibility*
>
> `A`: Thanks for the question. It is indeed possible that the same version of GPT-4 may exhibit slight performance variations over time due to OpenAI’s closed-source nature. However, our method does not overly rely on any specific capability unique to GPT-4, as might be the case in other tasks. Any large language model (LLM) with adequate language reasoning abilities can be used to implement our AgenticIR framework.
>
> To demonstrate this, we tested the performance of our method by replacing GPT-4 with the open-source Llama3-405B (referred to as AgenticIR (Llama)). The table below presents the results compared to the randomized and default settings of AgenticIR using GPT-4, as reported in the paper. The performance differences are minimal. This is because, with prior experience, the scheduling problem can be effectively handled even by a less capable LLM, and Llama produced results comparable to GPT-4. We believe that the minor performance differences are more likely due to the inherent randomness in tool invocation rather than model-specific capabilities.
> We will include this additional analysis and discussion in the revised version of the paper.
>
> | Degradations | Method            | PSNR  | SSIM   | LPIPS  | MANIQA | CLIPIQA | MUSIQ |
> |--------------|-------------------|-------|--------|--------|--------|---------|-------|
> | Group A      | Random            | 20.90 | 0.6642 | 0.3368 | 0.2963 | 0.4394  | 55.30 |
> | Group A      | AgenticIR         | 21.04 | 0.6818 | 0.3148 | 0.3071 | 0.4474  | 56.88 |
> | Group A      | AgenticIR (Llama) | 21.06 | 0.6834 | 0.3084 | 0.3123 | 0.4516  | 57.61 |
> | Group B      | Random            | 20.06 | 0.6766 | 0.3351 | 0.3120 | 0.4514  | 56.15 |
> | Group B      | AgenticIR         | 20.55 | 0.7009 | 0.3072 | 0.3204 | 0.4648  | 57.57 |
> | Group B      | AgenticIR (Llama) | 20.79 | 0.7019 | 0.3062 | 0.3174 | 0.4648  | 57.47 |
> | Group C      | Random            | 18.87 | 0.5456 | 0.4796 | 0.2354 | 0.3543  | 44.61 |
> | Group C      | AgenticIR         | 18.82 | 0.5474 | 0.4493 | 0.2698 | 0.3948  | 48.68 |
> | Group C      | AgenticIR (Llama) | 18.80 | 0.5480 | 0.4562 | 0.2675 | 0.3859  | 48.13 |

---

> ### Author Response · Authors · 2024-11-25
> **Follow-up discussions with Reviewer wMVY**
>
> Dear Reviewer wMVY,
>
> Thank you once again for your valuable time and insightful comments on our manuscript.
>
> We have provided detailed responses to your concerns, which we believe address all the issues you raised. We would greatly appreciate the opportunity to discuss with you whether our responses have satisfactorily resolved your concerns. Please let us know if there are any aspects of our work that remain unclear.
>
> We understand that your time is valuable, and we would be grateful if you could review our responses and share your thoughts at your earliest convenience. Please know that the opportunity for discussion is limited, so your timely feedback is greatly appreciated.
>
> Thank you for your consideration.
>
> Best regards,
>
> Authors

---

> ### Author Response · Authors · 2024-11-30
> **Respectful reminder for discussion**
>
> Dear Reviewer wMVY,
>
> The time remaining for discussion is running out. We are really looking forward to your feedback, which will be helpful for refining our work. If you find our response not addressing all your concerns, we are more than willing to provide further clarification. Once again, thank you for your valuable time and comments.
>
> Best regards,
>
> Authors

---

> ### Author Response · Authors · 2024-12-02
> **Final Follow-Up Discussion with Reviewer wMVY – Less Than 24 Hours Remaining**
>
> Dear Reviewer wMVY,
>
> Thank you for your valuable comments. We have carefully addressed the concerns you raised and provided detailed responses.
>
> The discussion period between authors and reviewers is about to conclude, **with approximately 24 hours remaining**. We have made multiple attempts to engage with you, and the AC has also issued two calls for discussion. However, we have yet to receive your response.
>
> **We believe that participating in the discussion before the deadline is both critical to ensuring clarity and a reflection of professional courtesy.**
>
> Could you kindly provide your feedback at your earliest convenience? Your time and input are greatly appreciated.
>
> Best regards,
>
> Authors

---

### Official Review · Reviewer_oGv6 · 2024-11-04

**Soundness:** 4
**Presentation:** 4
**Contribution:** 3
**Rating:** 6
**Confidence:** 5

**Summary:**

This paper presents an agentic workflow based on LLM/VLMs for image restoration. The agentic system follows how actual humans would process images, consisting of five stages: Perception, Scheduling, Execution, Reflection, and Rescheduling. Since the existing VLMs are not sufficiently capable of analyzing image quality or reasoning about the order of image processing tasks, the VLMs are finetuned and allowed for (self-)exploration to understand the effects of scheduling the image restoration components. Experimental results clearly demonstrate the effects of scheduling with learned experience and the other proposed components.

**Strengths:**

1. Clear presentation of the benefits of the proposed methodologies. I especially liked Figure 3, 6, 7, and 8, where the authors show dramatic improvements on why choosing a good scheduler for the image restoration components is important, as well as reflection and rollback.

2. Novelty in connecting the human process of IR with LLM-based agentic systems. Though the idea of mimicking the human workflow is being widely adopted more recently, application to image restoration tasks and showing effectiveness is not demonstrated before, to my knowledge.

3. Thorough justification of the design choices and careful experiment designs. Reasons for the proposed workflow and the capabilities the authors are trying to give to the LLM/VLMs are well described, and the evaluations seem to be fairly performed.

**Weaknesses:**

1. No cost analysis. Using such agentic systems require numerous requests to the LLM/VLM APIs; if the system chooses to perform "Reflection" with the tree search, the worst case scenario would be extremely costly. Compared to the existing image restoration models, the proposed model uses significantly more compute. In this sense, given that many previous works (roughly) match the FLOPs when comparing the restoration quality, one might argue that the comparisons are unfair.

2. Relatively subtle improvements for quantitative metrics (though qualitative improvements look quite significant). I would suggest also adding quantitative measures on the figures so that the readers can compare both aspects with a single glance.

**Questions:**

1. How are the discovered workflows similar to the original motivation of following the human workflow? For instance, is the subtask scheduling by GPT-4 (w. experience) match the best practices performed by a human? It would be better if the authors could provide more insights or discussions.

2. How does the proposed model perform when there is only a single type of degradation? Does it also perform competitively?

3. What is the criterion for deciding whether Execution step is Failed or Successful? (I might have missed)

---

> ### Author Response · Authors · 2024-11-21
> **Response to Reviewer oGv6**
>
> We sincerely thank the reviewer for the comments and recognition of the clarity and novelty of our work, as well as the thorough justification of our design choices. We especially appreciate the positive feedback on our figures and the effectiveness of scheduling, reflection, and rollback in the proposed methodologies.
>
> `1.` *Cost analysis*
>
> `A`: Thank you for the reviewer’s questions. We encourage the reviewer to refer to the public response at: https://openreview.net/forum?id=3RLxccFPHz&noteId=XtMKDGs0G7, where we have provided detailed explanations regarding efficiency, cost, and fairness concerns.
>
> `2.` *Relatively subtle improvements for quantitative metrics*
>
> `A`: Thanks for the question. The issue of evaluation metrics in image restoration has been a longstanding challenge in the field. In many cases, image quality metrics are insufficient in accurately reflecting perceived image quality, as demonstrated by numerous studies [R4, R5]. For example, two entirely different images—one slightly blurry and another with enhanced contrast—may yield very similar PSNR values. However, it is evident that the image with improved contrast offers a much better visual experience.
>
> Aware of the limitations of these metrics, we included up to six different metrics in our study to provide a more comprehensive evaluation of our method’s effectiveness. Our approach shows clear advantages across most metrics. In response to the reviewer’s concerns and suggestions, we will enhance our paper by including quantitative metrics in the figures to facilitate direct comparisons. Additionally, we will provide more comparative examples to offer a well-rounded demonstration of our method’s performance.
>
> `3.` *How are the discovered workflows similar to the original motivation of following the human workflow?*
>
> `A`: Thank you for the insightful question. Comparing our method to human operators is an excellent idea, and we are exploring ways to incorporate such discussions into our work. However, conducting such a study is inherently challenging due to the nature of human factors.
>
> First, the selection of human participants is crucial. If the participants are domain experts, it is difficult to find a sufficiently large and unbiased sample. On the other hand, using non-expert participants often results in highly inconsistent performance, influenced by factors such as educational background, patience, number of attempts, environment, and even mood. Additionally, our study is conducted in a controlled laboratory setting, which differs significantly from real-world scenarios. Introducing human participants under such conditions further complicates the comparison, as their actions would inevitably be constrained. That said, we believe this is an important perspective, and we are actively exploring feasible approaches to compare our method against human performance. These discussions will be included in future versions of our paper and in follow-up work on image processing agents.
>
> `4.` *How does the proposed model perform when there is only a single type of degradation?*
>
> `A`: Thank you for the insightful question. If the VLM identifies only one type of degradation, the AgenticIR simply selects the specialized tool designed for that specific degradation, and the final output will be the result of a single IR model deemed by the VLM to successfully address the issue. If all tools are considered failures, the VLM compares the outputs of all tools and selects the best one as a compromise for the final result.
>
> This represents a trivial case that does not fully showcase the capabilities of our method. However, if such a comparison must be made, we believe our approach remains competitive. Compared to unified models, our method leverages specialized models tailored for specific degradations, which are widely regarded as superior for such tasks [R1, R2, R3]. When compared to dedicated models (though this is an unfair comparison, as degradation information is leaked), our approach, through the reflection mechanism, can at least avoid the worst-case results. In fact, the more severe the degradation, the more effective our method becomes, as it can reject poor tool outputs and dynamically adapt to achieve better results.
>
> `5.` *What is the criterion for deciding whether Execution step is Failed or Successful?*
>
> `A`: Thank you for your question. During implementation, the Execution and Reflection stages are interleaved. As shown in Figure 2(c) and Appendix A.1, the agent randomly selects a tool to restore the image and then uses the VLM to reflect on the result. If the VLM determines that the tool has successfully addressed the degradation issue, the execution step concludes successfully. Otherwise, the agent continues to try other tools, repeating this process. If the VLM deems all available tools unsuccessful, the execution step concludes as a failure.

---

> > ### Author Response · Authors · 2024-11-21
> > **References mentioned in the response to reviewer oGv6**
> >
> > [R1] Shuo Cao, Yihao Liu, Wenlong Zhang, Yu Qiao, and Chao Dong. GRIDS: Grouped multiple-degradation restoration with image degradation similarity. In ECCV, 2024.
> >
> > [R2] Haoyu Chen, Wenbo Li, Jinjin Gu, Jingjing Ren, Sixiang Chen, Tian Ye, Renjing Pei, Kaiwen Zhou, Fenglong Song, and Lei Zhu. RestoreAgent: Autonomous image restoration agent via multimodal large language models. In NeurIPS, 2024.
> >
> > [R3] Zhang, Ruofan, Jinjin Gu, Haoyu Chen, Chao Dong, Yulun Zhang, and Wenming Yang. “Crafting training degradation distribution for the accuracy-generalization trade-off in real-world super-resolution.” In /International conference on machine learning/, pp. 41078-41091. PMLR, 2023.
> >
> > [R4] Yu, Fanghua, Jinjin Gu, Zheyuan Li, Jinfan Hu, Xiangtao Kong, Xintao Wang, Jingwen He, Yu Qiao, and Chao Dong. “Scaling up to excellence: Practicing model scaling for photo-realistic image restoration in the wild.” In /Proceedings of the IEEE/CVF Conference on Computer Vision and Pattern Recognition/, pp. 25669-25680. 2024.
> >
> > [R5] Jinjin, Gu, Cai Haoming, Chen Haoyu, Ye Xiaoxing, Jimmy S. Ren, and Dong Chao. “Pipal: a large-scale image quality assessment dataset for perceptual image restoration.” In /Computer Vision–ECCV 2020: 16th European Conference, Glasgow, UK, August 23–28, 2020, Proceedings, Part XI 16/, pp. 633-651. Springer International Publishing, 2020.

---

> > > ### Comment · Area_Chair_sZSc · 2024-11-24
> > > **Discussion Period Ending Soon**
> > >
> > > Dear Reviewer,
> > >
> > > The discussion period will end soon. Please take a look at the author's comments and begin a discussion.
> > >
> > > Thanks, Your AC

---

> > > > ### Comment · Area_Chair_sZSc · 2024-12-01
> > > > **Discuss**
> > > >
> > > > Dear Reviewer,
> > > >
> > > > Discussion is an important part of the review process. Please discuss the paper with the authors.
> > > >
> > > > Thanks, Your AC

---

> > ### Comment · Reviewer_oGv6 · 2024-12-02
> > **Response to Authors for the comments**
> >
> > Thanks for the detailed response. I understand that this paper provides a new approach that requires a lot of new design choices and that the authors were thoughtful in choosing the options for realizing the proposed method. Although there still remains a lot of rooms for improvement and optimization (especially on the engineering aspects, compared with the existing "well-developed" methods), I think this paper is generally well written and presented, and I'm still leaning towards accept.

---

### Author Response · Authors · 2024-11-20
**Response to Reviewers and ACs: Efficiency, Cost, and Fairness**

We sincerely thank all reviewers for their time and appreciation of our work. We are delighted that **the reviewers have recognized our paper’s novelty, presentation quality, and experimental rigor**. Before addressing individual reviewer comments, we would like to respond to several common concerns shared across the reviews.

Our work pioneered the introduction of an Agent perspective to image processing, receiving unanimous recognition from reviewers. **This novel paradigm represents a fundamental departure from traditional image processing methods and demonstrates immense potential.** As icebreaking research, we had to define problem frameworks and experimental environments from scratch, inevitably resulting in some exploratory limitations. However, these limitations precisely underscore the research value of this direction. Combined with the flourishing field of Agent research, we believe these challenges should not be grounds for rejecting this work, but rather serve as motivation for the academic community to further explore this promising new paradigm.

`Efficiency and Cost`

Our method indeed requires a relatively long inference time in many cases. To quantify the cost, we recorded the wall-clock time and the number of tool calls during each inference. The table below presents the average results for each group of experiments. On average, it takes about one minute to restore an image. The inference time consists of three main components: calls to the LLM (GPT-4), calls to the VLM (a fine-tuned DepictQA), and execution of the IR model. Among these, the first two components contribute relatively little to the overall time. The LLM is typically called only once for scheduling and, in a small number of cases (approximately 20%), may be called a second time for rescheduling. Experiments show that each LLM call takes less than five seconds on average. Calls to the VLM are primarily triggered during perception and reflection stages, with each call taking less than one second due to the brevity of the dialogues. Therefore, the main source of time and complexity comes from tool execution. Overall, the total time consumed by our method is roughly equivalent to the execution time of a single IR model multiplied by the number of tool calls.

| **Degradations** | **Wall clock time (s)** | **# Tool invocations** | **# Tool invocations / # Degradations** |
|---|---|---|---|
| Group A | 48 | 3.37 | 1.685 |
| Group B | 54 | 3.63 | 1.815 |
| Group C | 78 | 4.77 | 1.59 |

It is worth noting that the combined use of multiple tools to address complex image restoration tasks inherently requires the collaboration of multiple IR models. Even for humans, it would be challenging to significantly improve efficiency in similar tasks. If we compare AgenticIR to a human assistant, its time consumption is quite acceptable. Intelligence, by its nature, often comes at the cost of complexity. As the saying goes, "There’s no such thing as a free lunch." **High-level intelligent services inevitably come with certain computational overhead.**

The primary cost difference between our approach and previous methods lies in the use of large language models. Running large language models effectively does require significant computational resources. In our research, we primarily use publicly accessible language model APIs (e.g., GPT-4, LLaMa). **Generally, cutting-edge methods and tools are often less mature and efficient compared to widely adopted paradigms.** When deep learning first emerged, training a deep network required far more cost and time than traditional computer vision methods. Similarly, when large language models were initially introduced, their demand for computational resources was almost unimaginable to most practitioners at the time. However, as these technologies evolve, they gradually become more accessible and affordable. **We hope that reviewers can focus on the innovative intelligence and future potential of our method** rather than excessively critiquing its efficiency and cost-effectiveness at this stage. Issues related to efficiency and cost will naturally be addressed through continued research and development over time.

`Fair comparison`

We want to begin by emphasizing that we highly value fairness in comparisons. However, our method introduces a fundamentally different and more intelligent paradigm, one that consciously trades some simplicity for greater intelligence. When a research paradigm undergoes such a significant shift, achieving perfectly fair comparisons with traditional methods becomes inherently challenging. Nonetheless, we believe that the potential for enhanced intelligence offered by this new paradigm is far more compelling and deserving of exploration.

---

### Meta-Review · Area_Chair_sZSc · 2024-12-21

**Metareview:**

The paper addresses the problem of image restoration and proposes an agential system where a VLM and LLM can mimic a human workflow for image restoration (e.g. using specific image restoration tools, make decisions as to what tools should be used, reflecting on the current performance, etc.). The paper specifically proposes a 5 stage process composed of perception, scheduling, execution, reflection, and rescheduling in which the VLM attempts to: perceive specific degradation, make a plan as to how to address them (as well as the order in which to address them), apply the tools and then enter a loop of rescheduling, execution, and reflection where the VLM can decide whether a step succeeded or not and how to proceed from there.

The main strength of the paper is its interesting proposed method and strong qualitative results as well as decent quantitative results. While agentic workflows are becoming more commonplace, a strong demonstration of how they can apply to vision (e.g. image restoration) is very interesting and useful for the field. The main weakness seems to be inference time. However as this is, to my and the reviewers' knowledge, the first attempt at an agential workflow, I value the interestingness of the approach and the performance more.

I advocate for acceptance.

**Additional Comments On Reviewer Discussion:**

Reviewer wMVY rates the paper a 5 raising concerns about the comparison fairness, efficiency, the lack of a toolbox ablation, and the usage of GPT4. While I agree that a toolbox ablation would improve the paper, I think the comparison fairness, efficiency, and lack of GPT4 seem relatively minor compared to the strengths of the paper. The reviewer did not participate during the rebuttal period.

Reviewers oGv6, sgx9 rate the paper a 6. Reviewer hoAk rates the paper an 8. In general, I agree with their assessment which indicate that the proposed scheme is interesting and of value to the community.

---

### Decision · Program_Chairs · 2025-01-22

Accept (Poster)